

Significant contributions of biomass burning to PM$_{2.5}$-bound aromatic
compounds: insights from field observations and quantum chemical
calculations
Yanqin Ren[1], Zhenhai Wu[1], Fang Bi[1], Hong Li[1], Haijie Zhang[1*], Junling Li[1], Rui Gao[1],
Fangyun Long[1], Zhengyang Liu[1], Yuanyuan Ji[1*], Gehui Wang[2*]
[1] State Key Laboratory of Environmental Criteria and Risk Assessment, Chinese
Research Academy of Environmental Sciences, Beijing 100012, China
[2] Key Lab of Geographic Information Science of Ministry of Education of China,
School of Geographic Sciences, East China Normal University, Shanghai 200142,
China
*Corresponding authors: Dr. Haijie Zhang /Dr. Yuanyuan Ji/ Prof. Gehui Wang
E-mail address: zhanghaijie@craes.org.cn / ji.yuanyuan@craes.org.cn
/ghwang@geo.ecnu.edu.cn



**Abstract**

20        Polycyclic aromatic hydrocarbons (PAHs), oxygenated PAHs (OPAHs), and

nitrated phenols (NPs) are essential aromatic compounds that significantly affect both
climate and human health. However, their sources and formation mechanisms,
particularly for NPs, remain poorly understood. This study determined the
concentration profiles and the main formation mechanisms of these substance classes
in $PM_{2.5}$ from Dongying, based on field observations and quantum chemical
calculations. The daily concentrations of ∑13PAHs during heating were more than
twice higher compared to those before the heating period. Benzo($b$)fluoranthene was
identified as the primary PAHs species. The average concentration of ∑8OPAHs
reached 351 ng m⁻³, with significantly increased concentrations observed during the
heating season, and 1-Naphthaldehyde (1-NapA) emerged as the most prevalent OPAH
species. Concentrations of ∑9NPs increased approximately 1.2 times during the heating,
with 4-methyl-5-nitrocatechol (4M5NC) having the highest concentration. Positive
matrix factorization analysis identified biomass burning to be the primary source of
these aromatic compounds, particularly for PAHs. Density functional theory
calculations further revealed that phenol and nitrobenzene are two main primary
precursors for 4-nitrophenol, with phenol showing lower reaction barriers, and $P$-Cresol
was identified as the primary precursor for the formation of 4M5NC. This study
provides the first detailed investigation of the sources and formation mechanisms of
aromatic compounds in the atmosphere of petrochemical cities in the Yellow River
Delta, which may provide fundamental insights and important guidance for reducing



emissions of aromatic compounds in similar atmospheric environments.

**Keywords:** Aromatic compounds, Source identification, DFT calculations, Formation
mechanism, Heating period
**1. Introduction**

Aromatic compounds, characterized by the presence benzene ring, are known for

their structural stability and resistance to decomposition, as well as their distinctive
aromatic properties. Polycyclic aromatic hydrocarbons (PAHs), oxygenated PAHs
(OPAHs), and nitrated phenols (NPs) are significant aromatic compounds commonly
found in atmospheric particulate matter. These compounds exerted a considerable
influence on ambient air quality, climate change, and human health (Peng et al., 2023;
Yhab et al., 2021; Chong et al., 2021; Lammel et al., 2020; Elzein et al., 2019).

PAHs are semi-volatile compounds, comprised of multiple interconnected

aromatic rings and are commonly found throughout various environmental settings.
PAHs are primarily formed as byproducts of the partial combustion of carbon-rich
substances, including coal, biomass, tobacco, garbage, charbroiled meat, and petroleum
(Del Rosario Sienra et al., 2005; Kashiwakura and Sakamoto, 2010; Chen et al., 2005;
Shen et al., 2010). OPAHs are aromatic compounds with one or more carbonyl groups
connected to their ring structure, and they contain various quinones and ketones.
OPAHs generally have lower vapor pressures upon comparison with their parent PAHs,
leading to a higher propensity to be retained in the particulate phase. The direct-acting



mutagen properties and the generation of reactive $O_2$ species render some OPAHs more
hazardous compared with the parent PAHs (Bandowe et al., 2010; Chung et al., 2006;
Bolton et al., 2000; Pedersen et al., 2005). OPAHs is carried out either directly through
the incomplete combustion of various organic materials (Oda et al., 2001; Jakober et
al., 2007) or can be formed indirectly through photochemical reactions involving PAHs
and ozone, nitro, and hydroxyl radicals (Bandowe et al., 2014; Wang et al., 2011; Lin
et al., 2015b). Research suggests that the PAH concentrations, along with their
derivatives in $PM_{2.5}$, increased during winter or heating periods, primarily due to
biomass and coal burning (Lin et al., 2015b). Furthermore, particulate matter released
from biomass burning is considered more toxic compared to that derived from other
sources (Sarigiannis et al., 2015). NPs are monocyclic aromatic compounds having
properties similar to benzene, including low solubility in water while high solubility in
various organic solvents such as ethanol, pyridine, xylene, and chloroform. In addition
to adversely affecting human health, NPs can also disrupt the equilibrium of the
ecosystem, increase the risk of cancer, and disrupt plant growth (Chow et al., 2015;
Liang et al., 2020; Booth et al., 2014). NPs in the atmosphere are primarily derived
from primary emissions and secondary formation. Primary sources of NPs include
emissions from coal combustion, vehicle exhaust, biomass combustion, and industrial
and agricultural sources, similar to the sources of PAHs and OPAHs (Liang et al., 2020;
Wang et al., 2017; Lu et al., 2021). Observations in the atmospheric environment
indicate that secondary formation can provide more than one-third of the total NPs
present (Yuan et al., 2016a). Benzene, toluene, and their derivatives are considered





significant precursors in the secondary formation of NPs, although secondary NPs can
also be generated through nitrification of phenols in the gas or condensation phase or
by interactions between phenoxyl radicals, formed from $NO_2$, and other aromatic
compounds (Harrison et al., 2005).

In Dongying City, known for its petrochemical industry, benzene, toluene, and

xylene are significant contributors to the levels of VOCs into the environment (Chen et
al., 2020). Moreover, research has revealed that in the presence of $NO_x$, aromatic
hydrocarbons can be oxidized to yield nitroaromatic compounds, including nitrophenol,
dinitrophenol, nitrocatechol, and methylnitrocatechol (Lin et al., 2015a). The primary
sources of aromatic compounds in particulate matter within high-aromatic
environments and the potential relationship between their secondary formation and the
benzene series mentioned earlier still require systematic investigation. Therefore, this
study explored the pollution characteristics and primary sources of PAHs, OPAHs, and
NPs in $PM_{2.5}$ in Dongying City, and further explored the formation mechanism of
typical nitrophenols based on field observations and density functional theory
calculations. This study represents the first investigation regarding the sources and the
forming mechanisms of aromatic compounds within the atmosphere of petrochemical
cities in the Yellow River Delta.
**2 Materials and Methods**
**2.1 Field observations**

$PM_{2.5}$ samples collection was carried out at Dongying Atmospheric Super Station



(118.59°E, 37.45°N) from October 27 to December 6, 2021. The surrounding area is
primarily residential and commercial, with well-developed transportation infrastructure
and no significant sources of industrial pollution, making it a representative urban
monitoring site. A high-flow particulate sampler (TH-1000H, Wuhan Tianhong
Instrument Co., LTD) was employed to obtain the samples. This instrument had a
sampling flow rate equal to 1.05 m³/min and a flow accuracy of ±2.5%. It comprised a
quartz filter (Quartz, 203 mm×254 mm, Whatman, UK), with an effective filter
diameter of 180 mm×230 mm. Samples were collected twice daily: once during the day
from 8:00 am to 7:30 pm local time (LT) and once from 8:00 pm to 7:30 am LT.
Meteorological statistics, including temperature, weather conditions, humidity, wind
direction and speed, were measured concurrently with each sampling.
**2.2 Chemical analysis**

Organic species in $PM_{2.5}$ were detected following a pre-treatment process that

included ultrasonic extraction along with derivatization. Initially, a quarter of the
membrane was placed into a sample bottle and then immersed in a mixture of
dichloromethane and methanol solution (2:1, v/v) to completely submerge the filters.
After that, three 15-min treatments of ultrasonic extraction were carried out. Following
the process of extraction, filtration was carried out through glass wool using a Babbitt
dropper into pear-shaped flasks. The filtrates were then concentrated to a small amount
through a rotary evaporator inside a vacuum. Subsequently, they were transferred to
GC bottles, which were dried with a nitrogen purifier, followed by the addition of 60
µL of N, O-bis-(trimethylsilyl) trifluoroacetamide solution. The bottles were then





placed in an oven at 70 °C for a total time duration of 3 h to complete the derivatization
process. After the reaction, the solution was cooled and an internal standard comprised
of 40 µL tridecane was added. The solution was mixed completely and then placed in
the refrigerator for examination.

This study analyzed thirteen different PAHs, eight different OPAHs, and nine

different NPs using the afore mentioned methods of analysis, as well as some organic
tracers, e.g. levoglucosan. The thirteen kinds of PAHs include Fluoranthene (Flu),
Benzo[k]fluoranthene   (BkF),   Benzo[a]anthracene   (BaA),   Pyrene   (Pyr),
Chrysene/triphenylene(CT), Benzo[b]fluoranthene (BbF), Benzo[a]pyrene (BaP),
Benzo[e]pyrene (BeP), Indeno[1,2,3-cd] (IP), Perylene (Per), Benzo[ghi]perylene
(BghiP), Dibenzo[a,h]anthracene (DBA), and Coronene. The eight OPAHs include
benzanthrone (BZA), 9-fluorenone (9-FO), 1-Naphthaldehyde (1-NapA), 1,4-
chrysenequione (1,4-CQ), anthraquinone (ATQ), 5,12-naphthacenequione (5,12-NAQ),
benzo(a)anthracene-7,12-dione   (7,12-BaAQ),   and   6H-benzo(cd)pyrene-6-one
(BPYRone). The nine NPs are 4-nitrophenol (4NP), 4-nitroguaiacol (4NGA), 3-methyl-
4-nitrophenol (3M4NC), 4-methyl-5-nitrocatechol (4M5NC), 4-nitrocatechol (4NC),
5-nitroguaiacol (5NGA), 2, 4-dinitrophenol (2, 4-DNP), 5-nitro-salicylic acid (5NSA)
and 3-nitro-salicylic acid (3NSA).

Inorganic ions soluble in water were quantified through an ion chromatograph

(Dionex-1100). The analysis focused on eight water-soluble inorganic ions: $NO_3^-$, $Cl^-$,
$NH_4^+$, $SO_4^{2-}$, $Na^+$, $Mg^{2+}$, $K^+$, and $Ca^{2+}$. The measurement of organic carbon (OC) and
elemental carbon (EC) was performed through a thermal/optical carbon analyzer





(DRI2015). Detailed tests for analyzing EC, OC, and inorganic ions are provided in a
previous report (Ren et al., 2021).
**2.3 Quantum chemical calculations**
Quantum chemical calculations both in the gas phase and liquid phase were
performed using the Gaussian 09 software. The geometries and frequencies for the
reactant monomers, reactant complexes (RC), transition states (TS), intermediate (IM)
and product complexes (PC) were performed at the M06-2X/6-311++G(2df,2p) level
of theory,  which is well-established for studies of organic systems (Li and Wang,
2014; Zhao and Truhlar, 2007). The conductor-like polarizable continuum
model(CPCM) was used for the optimization calculations to include the solvent effect
in the liquid (Takano and Houk, 2005). Radicals were treated using unrestricted
Hartree–Fock calculations, and TS was optimized to ensure the presence of a single
imaginary frequency. Intrinsic reaction coordinate (IRC) calculations were carried out
to further guarantee that the TS connects the right pre- and post-complexes along the
reaction coordinate (Gonzalez and Schlegel, 1989).
**2.4 Quality control and quality assurance (QC&QA)**
QA and QC procedures for sampling and laboratory analysis strictly follow the
environmental monitoring technical specifications and analytical standards established
by the Ministry of Environmental Protection. Periodic inspections of all analytical and
testing instruments were carried out in accordance with the relevant metrological
verification protocols. The quartz filter membrane was burned inside a Muffle furnace
at a temperature of 450 ℃ for 6 h before sampling. After natural cooling, it was



removed and transferred to a controlled temperature and humidity box ($25 \pm 1$ °C, $50 \pm$
5% RH). The sampled filter membrane was packed in a ziplock bag and stored in the
refrigerator at a temperature below 0 °C until analysis. To minimize systematic errors,
corresponding blank filter samples were simultaneously prepared and stored followed
by analysis under the same conditions as the collected samples.
**3 Results and discussion**
**3.1 Changes in meteorological conditions, gaseous pollution, and major**
**components of $PM_{2.5}$**

The entire study period was divided into two stages: before heating (October 27 to

November 14, 2021) and during heating (November 15 to December 6, 2021). Fig. 1
and Table 1 illustrate the temporal variations within the meteorological factors,
concentrations of gaseous pollutants, as well as the primary components of $PM_{2.5}$.
Before heating, the relative humidity (RH) and temperature (T) were higher, averaging
$64 \pm 18\%$ and $12 \pm 5.4$ °C, respectively. During the heating period, the average values
were found to be lower at $60 \pm 18\%$ and $7.9 \pm 3.7$ °C. Overall, the average values for
the entire study period were determined to be $62 \pm 18\%$ and $9.6 \pm 4.9$ °C. The
concentrations of $SO_2$ and NO during heating ($11 \pm 6.6$ ppb, and $17 \pm 16$ ppb,
respectively) were nearly twice that of the value observed before heating, with a slight
increase in the level of $NO_2$. These increases are likely attributable to the increased
burning activities during the heating season, particularly residential coal combustion.
In contrast, the levels of $O_3$ and $O_X$ (the sum of $NO_2$ and $O_3$) were significantly higher



before heating, with an average value of $49 \pm 27$ ppb and $80 \pm 28$ ppb, respectively, in
comparison to that observed during the heating period. This variation was primarily
ascribed to the higher temperatures and stronger solar radiation before heating.
Although the $PM_{2.5}$ concentrations remained unchanged throughout the sampling
period, there were considerable variations in the concentrations and proportions of its
chemical components. Figs.1 and 2 showed the time variations and absolute proportion
of chemical components in $PM_{2.5}$ before and during heating, respectively, throughout
the entire sampling period. Throughout the observation period, secondary inorganic
aerosols (SIA) such as $SO_4^{2-}$, $NO_3^-$, and $NH_4^+$ were determined as the main chemical
component of $PM_{2.5}$, followed by OM (1.6 times the OC). Before heating, the
concentrations of $SO_4^{2-}$, $NO_3^-$, and $NH_4^+$ were found to be 5.5, 17, and 7.5 µg m$^{-3}$,
respectively, while during heating, they demonstrated the average values of 5.3, 17, and
7.7 µg m$^{-3}$ (Fig. 1b, c). The total concentrations and proportions of SIA remained
relatively stable, with an average of about 35% during the whole process. Among these,
$NO_3^-$ was determined as the predominant species, constituting approximately 19% of
the total $PM_{2.5}$ during the sampling period. OM comprised 12.7% of $PM_{2.5}$ before the
heating period and increased significantly to 17.5% during heating (Fig. 2b, c).
Correspondingly, the average OC concentration increased from 5.4 to 7.7 µg m$^{-3}$.
However, the average ratio of OC/EC remained relatively unchanged, with values of
$2.5 \pm 1.1$ before heating and $2.4 \pm 0.6$ during heating. Generally, OC is generated from
both primary emissions and secondary formation, while EC is predominantly derived
from primary sources. The significant increase in the concentration of OC alongside a



stable OC/EC ratio suggests that the sources of carbonaceous aerosols in this study
include both primary emissions and secondary formation processes. Furthermore, a
high positive correlation between levoglucosan and $K^+$ and $Cl^-$ (Fig. 3), along with the
enhanced mass concentrations of these ions during the heating period, suggests
additional combustion activities, including biomass burning, in addition to coal
combustion.

Table 1. Meteorological parameter values and concentrations of gaseous pollution and chemical components of PM2.5 during the sampling periods in Dongying.

| Period | The whole sampling | Before heating period | During heating period |
|---|---|---|---|
| Date | 27/10–6/12 | 27/10–14/11 | 15/11–6/12 |
| Scope of the Sample | $N$=74 | $N$=30 | $N$=44 |
| **Meteorological parameters** | | | |
| Temperature, ℃ | 9.6 ± 4.9 ((-1.7) – 21) | 12 ± 5.4 ((-1.7) – 21) | 7.9 ± 3.7 (0.2 – 17) |
| Relative humidity, % | 62 ± 18 (29 – 94) | 64 ± 18 (33 – 95) | 60 ± 18 (30 – 94) |
| **Gaseous pollutants, ppb** | | | |
| $SO_2$ | 8.9 ± 5.9 (2.7 – 28) | 6.7 ± 4.2 (3.1 – 20) | 11 ± 6.6 (2.7 – 28) |
| NO | 13 ± 14 (NA – 54) | 8.5 ± 11 (0.2 – 41) | 17 ± 16 (1.0 – 54) |
| $NO_2$ | 35 ± 17 (6.8 – 78) | 32 ± 18 (6.8 – 78) | 38 ± 16 (9.2 – 73) |
| $O_3$ | 39 ± 23 (3.8 – 106) | 49 ± 27 (3.8 – 106) | 30 ± 16 (8.9 – 83) |
| $O_X$ | 74 ± 25 (43 – 167) | 80 ± 28 (51 – 167) | 68 ± 20 (43 – 137) |
| **Major components of PM$_{2.5}$, µg m$^{-3}$** | | | |
| PM$_{2.5}$ | 74 ± 42 (23 – 222) | 74 ± 41 (23 – 167) | 75 ± 43 (25 – 222) |
| OC | 6.8 ± 3.5 (0.6 – 15) | 5.4 ± 2.7 (0.6 – 10) | 7.7 ± 3.8 (1.4 – 15) |
| EC | 3.7 ± 3.8 (0.2 – 26) | 3.5 ± 4.7 (0.2 – 26) | 3.8 ± 3.2 (0.6 – 20) |
| OC/EC | 2.4 ± 0.8 (0.1 – 5.1) | 2.5 ± 1.1 (0.1 – 5.1) | 2.4 ± 0.6 (0.2 – 3.6) |
| $SO_4^{2-}$ | 5.4 ± 3.5 (1.2 – 17) | 5.5 ± 3.6 (1.2 – 15) | 5.3 ± 3.4 (1.5 – 17) |
| $NO_3^-$ | 17 ± 15 (1.5 – 67) | 17 ± 15 (1.5 – 46) | 17 ± 16 (1.6 – 67) |
| $NH_4^+$ | 7.6 ± 6.1 (0.6 – 27) | 7.5 ± 6.1 (0.9 – 21) | 7.7 ± 6.2 (0.6 – 27) |
| $Cl^-$ | 1.8 ± 1.2 (0.2 – 6.1) | 1.5 ± 1.3 (0.2 – 6.1) | 2.0 ± 1.1 (0.5 – 5.3) |
| $K^+$ | 0.6 ± 0.4 (0.1 – 2.0) | 0.5 ± 0.4 (0.1 – 2.0) | 0.6 ± 0.3 (0.1 – 1.2) |







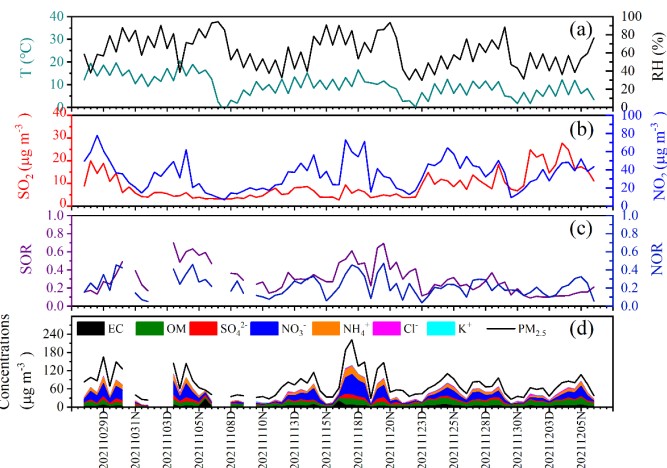

Fig.1 Time series of (a) T and RH, (b) $NO_2$ and $SO_2$, (c) SOR and NOR, and (d) $PM_{2.5}$ and its main compounds, in the winter and autumn of Dongying.

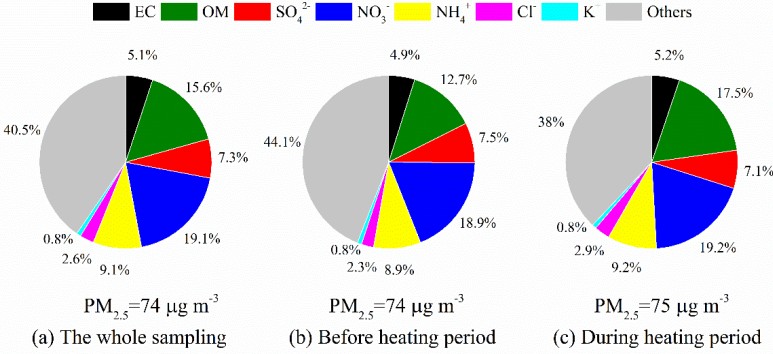

Fig.2 Absolute proportion of chemical components in $PM_{2.5}$ across the whole sampling period (a), before (b), and during the heating periods (c).

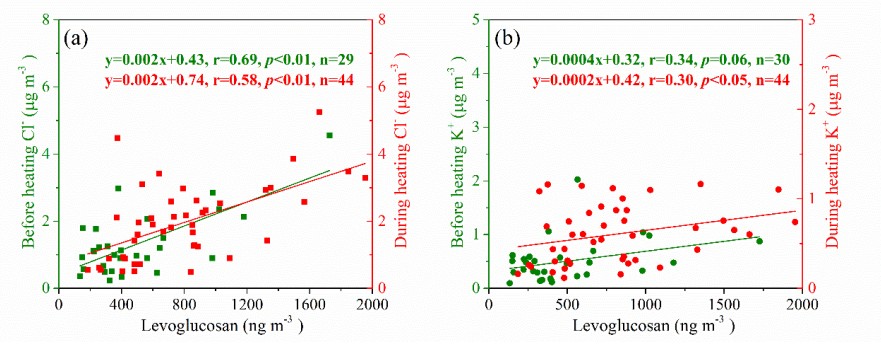

Fig.3 Regression analysis showing the linear relationship between levoglucosan and $Cl^-$ (a) and $K^+$ (b) before (green) and during (red) heating throughout the sampling period.



**3.2 Changes in the concentration and composition of aromatic compounds**

**3.2.1 PAHs and OPAHs**

Table 2 and Fig. 4b show that the daily concentrations of 13 different PAHs ($\sum$ 13PAHs) ranged from 3.9 to 388 ng·m$^{-3}$, with 91 ng·m$^{-3}$ of overall average value throughout the sampling period. During the heating phase, concentrations averaged 115 ng·m$^{-3}$ (over a range of 15 to 388 ng·m$^{-3}$), more than twice the levels recorded before heating, which averaged 56 ng·m$^{-3}$ (over a range of 3.9 to 241 ng·m$^{-3}$). The $\sum$13 PAHs measured in the current research were lower in comparison to those recorded in other Chinese cities, including Xi 'an (127 ng·m$^{-3}$, 2016–2017) (Wang et al., 2019a) and Harbin (215 ng·m$^{-3}$, 2017–2018) (Ma et al., 2020). Nighttime average concentrations of PAHs (65 ± 77 ng·m$^{-3}$ before heating and 154 ± 120 ng·m$^{-3}$ during heating) were observed to be 1.5 to 2 times higher upon comparison with those measured during the daytime (44 ± 39 ng·m$^{-3}$ before heating and 77 ± 49 ng·m$^{-3}$ during heating) (Fig. 5a). The observed day-night pattern aligns with the observations obtained from coastal cities in Bohai and the Yellow Seas, as well as Jinan in Shandong Province (Chen et al., 2021; Zhang et al., 2018). The concentration level of BbF in PAHs was found to be highest, accounting 16% of total PAHs (Fig. 4b), This dominance of BbF was found to be consistent with the previous research (Li et al., 2013; Ren et al., 2017; Bai et al., 2023; Li et al., 2022).

The average total concentration of OPAHs ($\sum$8OPAHs) was determined to be 351 ng·m$^{-3}$ across all the sampling periods, which was substantially higher during the heating period (average 378 ng·m$^{-3}$, range 114–812 ng·m$^{-3}$) compared to the period of





before heating (average 311 ng·m⁻³, range 70–875 ng·m⁻³) (Table 2). The concentration
of ∑8OPAHs was higher at night upon comparison with daytime, being approximately
1.1 to 1.2 times greater both before and during the heating period (Fig. 4c and 5b).
Among the eight OPAHs, 1-NapA was determined as the most prevalent, accounting
for 187 ng m⁻³ (59% of ∑8OPAHs) before heating and 156 ng m⁻³ (41% of ∑8OPAHs)
during heating (Fig. 4c and 5e). The average ∑8OPAHs concentrations in this study
were significantly higher compared with those found for other Chinese urban areas,
such as Guangzhou and Xi'an, with respective values of 23 and 54 ng m⁻³ (Ren et al.,
2017). Furthermore, the ∑8OPAHs levels observed in this study exceeded those
reported at various foreign sites, such as the south (traffic site, 41.8 ng m⁻³) (Alves et
al., 2017) and central European cities (about 10 ng m⁻³) (Lammel et al., 2020),
Thessaloniki, Greece (0.86–4.3 ng m⁻³) (Kitanovski et al., 2020), and Mainz, Germany
(0.047–1.6 ng m⁻³).
**3.2.2 NPs**

Based on the data presented in Table 2, the total concentrations of NPs compounds

(∑9NPs) demonstrated a mean value equal to 72 ng m⁻³ across the sampling period.
During the heating phase, ∑9NPs averaged 76 ng m⁻³ (covering a range extending from
23 to 175 ng m⁻³), which was approximately 1.2 times higher compared to the
concentrations determined before heating, with an average of 65 ng m⁻³ (covering a
range of 4.2 to 149 ng m⁻³). These concentrations were approximately twice those
observed during autumn and winter (38 ng m⁻³) in earlier work conducted in Beijing



(Ren et al., 2024). They were also considerably higher in comparison to the values
recorded in the summer (8.5 ng m$^{-3}$) and spring (8.6 ng m$^{-3}$) (Ren et al., 2022). In
contrast to OPAHs and PAHs, the total concentrations of ∑9NPs did not demonstrate
significant diurnal variation. However, during the heating period, a significant
nighttime increase was observed, with approximately 1.4 times higher concentrations
at night upon comparison with the levels of daytime (Fig. 4d and Fig. 5c). However,
the relative nine NPs molecular composition in PM$_{2.5}$ remained consistent throughout
the observation period. Among all the species, 4M5NC demonstrated the highest
concentration, contributing to 73% of the total NPs before heating and 53% during
heating, followed by 4NP, comprising 20% before heating and increased to 34% during
heating (Fig.4d and Fig. 5f). The total concentration of ∑9NPs observed in the current
research was generally comparable to previous measurements carried out in Beijing
during winter (74 ± 51 ng m$^{-3}$) (Li et al., 2020). Conversely, the levels were substantially
higher compared to those recorded in Jinan (48 ± 26 ng m$^{-3}$) (Wang et al., 2018), Hong
Kong, and Xi'an, with respective values of 12 ± 14 and 17 ± 12 ng m$^{-3}$ (Wu et al., 2020;
Chow et al., 2015). Compared to the international research, the observed ∑9NPs
concentrations were generally higher, such as those reported in Germany (16 ng m$^{-3}$),
the UK (19 ng m$^{-3}$), and Belgium (32 ng m$^{-3}$ in winter, 13 ng m$^{-3}$ in autumn) (Teich et
al., 2017; Mohr et al., 2013; Kahnt et al., 2013).

Table 2. Organic compounds in PM$_{2.5}$ during the sampling periods in Dongying.

| Period | The whole sampling | Before heating period | During heating period |
|---|---|---|---|
| Date | 27/10–6/12 | 27/10–14/11 | 15/11–6/12 |
| Scope of the Sample | *N*=74 | *N*=30 | *N*=44 |
| **PAHs** | | | |



| | | | |
|---|---|---|---|
| Fluoranthene (Flu) | 8.9 ± 8.1 (0.8 – 33) | 5.0 ± 4.3 (0.8 – 15) | 12 ± 9.0 (1.7 – 33) |
| Pyrene (Pyr) | 8.5 ± 8.4 (0.7 – 36) | 4.7 ± 4.3 (0.7 – 16) | 11 ± 9.5 (1.4 – 36) |
| benz(*a*)anthracene (BaA) | 6.8 ± 9.3 (0.1 – 47) | 3.5 ± 4.5 (0.1 – 20) | 9.0 ± 11 (0.7 – 47) |
| chrysene/triphenylene (CT) | 11 ± 12 (0.7 – 54) | 6.4 ± 6.7 (0.7 – 26) | 15 ± 13 (1.9 – 54) |
| benzo(*b*)fluoranthene (BbF) | 15 ± 15 (0.5 – 64) | 9.2 ± 10 (0.5 – 39) | 19 ± 16 (3.0 – 64) |
| benzo(*k*)fluoranthene (BkF) | 4.4 ± 4.0 (0.2 – 17) | 2.9 ± 2.8 (0.2 – 11) | 5.4 ± 4.4 (0.8 – 17) |
| benzo(*e*)pyrene (BeP) | 9.0 ± 8.5 (0.4 – 37) | 5.5 ± 6.1 (0.4 – 24) | 11 ± 9.1 (1.8 – 37) |
| benzo(*a*)pyrene (BaP) | 7.5 ± 8.7 (0.1 – 38) | 4.5 ± 6.0 (0.1 – 25) | 9.5 ± 9.7 (0.9 – 38) |
| Perylene (Per) | 1.6 ± 2.1 (0.01 – 10) | 0.9 ± 1.4 (0.01 – 57) | 2.0 ± 2.4 (0.2 – 9.6) |
| indeno(1,2,3-*cd*)pyrene (IP) | 7.6 ± 7.2 (0.1 – 26) | 5.5 ± 6.7 (0.01 – 24) | 8.8 ± 7.3 (0.8 – 26) |
| benzo(*ghi*)perylene (BghiP) | 6.8 ± 6.4 (0.1 – 23) | 5.0 ± 6.1 (0.02 – 23) | 7.9 ± 6.4 (0.9 – 23) |
| dibenz(*a,h*)anthracene (DBA) | 1.9 ± 2.4 (0.01 – 15) | 1.7 ± 2.1 (0.01 – 7.8) | 2.0 ± 2.5 (0.1 – 15) |
| coronene | 2.5 ± 2.6 (0.05 – 11) | 2.9 ± 3.6 (0.05 – 11) | 2.4 ± 2.2 (0.1 – 9.0) |
| Σ13PAHs | 91 ± 90 (3.9 – 388) | 56 ± 62 (3.9 – 241) | 115 ± 98 (15 – 388) |
| **OPAHs** | | | |
| 1-naphthaldehyde (1-NapA) | 168 ± 106 (57 – 796) | 187 ± 149 (57 – 796) | 156 ± 62 (75 – 320) |
| 9-fluorenone (9-FO) | 12 ± 12 (0.2 – 47) | 6.6 ± 7.1 (0.2 – 22) | 16 ± 13 (1.0 – 47) |
| anthraquinone (ATQ) | 64 ± 50 (1.1 – 282) | 37 ± 29 (1.1 – 103) | 83 ± 53 (10 – 282) |
| benzathrone (BZA) | 22 ± 22 (0.2 – 95) | 12 ± 15 (0.2 – 58) | 29 ± 24 (2.8 – 95) |
| benzo(a)anthracene-7,12-dione (7,12-BaAQ) | 6.3 ± 5.3 (0.5 – 24) | 3.8 ± 3.9 (0.5 – 16) | 7.9 ± 5.4 (1.3 – 24) |
| 1,4-chrysenequione (1,4-CQ) | 34 ± 34 (2.0 – 168) | 39 ± 41 (9.2 – 168) | 30 ± 28 (2.0 – 136) |
| 5,12-naphthacenequione (5,12-NAQ) | 2.6 ± 2.4 (0.02 – 9.4) | 1.6 ± 2.1 (0.2 – 8.1) | 3.2 ± 2.4 (0.5 – 9.4) |
| 6H-benzo(cd)pyrene-6-one (BPYRone) | 47 ± 46 (0.5 – 196) | 32 ± 45 (0.5 – 166) | 57 ± 45 (4.7 – 196) |
| Σ8OPAHs | 351 ± 196 (70 – 875) | 311 ± 209 (70 – 875) | 378 ± 185 (114 – 815) |
| **NPs** | | | |
| 4-nitrophenol (4NP) | 21 ± 20 (0.4 – 95) | 13 ± 16 (0.42 – 54) | 26 ± 21 (1.8 – 95) |
| 3-methyl-4-nitrophenol (3M4NP) | 2.0 ± 2.0 (0.04 – 7.8) | 1.0 ± 1.3 (0.04 – 4.8) | 2.6 ± 2.2 (0.16 – 7.8) |
| 4-nitroguaiacol (4NGA) | 1.0 ± 1.4 (0.02 – 7.5) | 0.5 ± 0.8 (0.02 – 2.9) | 1.4 ± 1.6 (0.04 – 7.5) |
| 5-nitroguaiacol (5NGA) | 0.09 ± 0.07 (0.01 – 0.3) | 0.12 ± 0.07 (0.01 – 0.3) | 0.07 ± 0.06 (0.01 – 0.22) |
| 4-nitrocatechol (4NC) | 4.1 ± 4.4 (0.1 – 21) | 2.6 ± 3.5 (0.1 – 16) | 5.0 ± 4.7 (0.39 – 21) |
| 2,4-dinitrophenol (2,4-DNP) | 0.24 ± 0.29 (0.004 – 1.3) | 0.14 ± 0.22 (0.004 – 0.92) | 0.31 ± 0.31 (0.006 – 1.3) |
| 4-methyl-5-nitrocatechol (4M5NC) | 48 ± 49 (8.7 – 385) | 49 ± 35 (8.7 – 125) | 40 ± 25 (8.9 – 116) |
| 3-nitrosalicylic acid (3NSA) | 0.03 ± 0.03 (NA – 0.17) | 0.03 ± 0.03 (0.003 – 0.17) | 0.03 ± 0.03 (0.01 – 0.13) |
| 3-nitrosalicylic acid (5NSA) | 0.63 ± 0.45 (0.07 – 1.8) | 0.48 ± 0.40 (0.07 – 1.8) | 0.72 ± 0.46 (0.07 – 1.8) |
| Σ9NPs | 72 ± 44 (4.2 – 175) | 65 ± 42 (4.2 – 149) | 76 ± 45 (23 – 175) |
| **Levonglucosan** | 679 ± 431 (134 – 1954) | 497 ± 362 (134 – 1727) | 803 ± 433 (185 – 1954) |




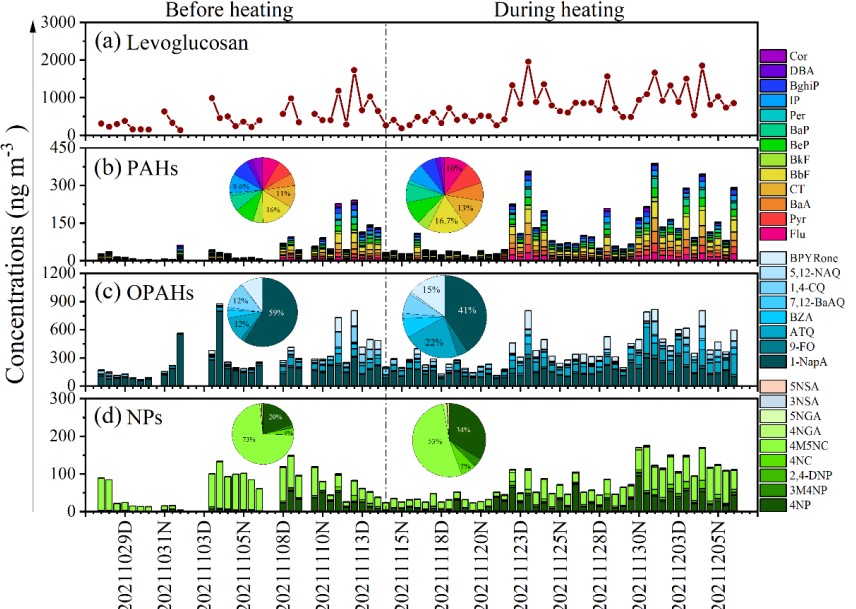


Fig.4 Time series of levoglucosan (a), PAHs(b), OPAHs (c), and NPs (d) in the
autumn and winter of Dongying.

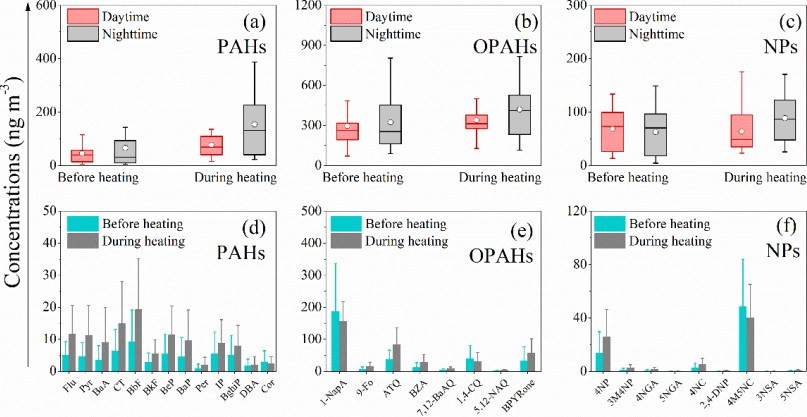


Fig.5 Diurnal variation and component concentrations of PAHs (a, d), OPAHs (b,
e), and NPs (c, f) before and during heating. Markers represent mean values, while
whiskers denote the 25th and 75th percentiles.

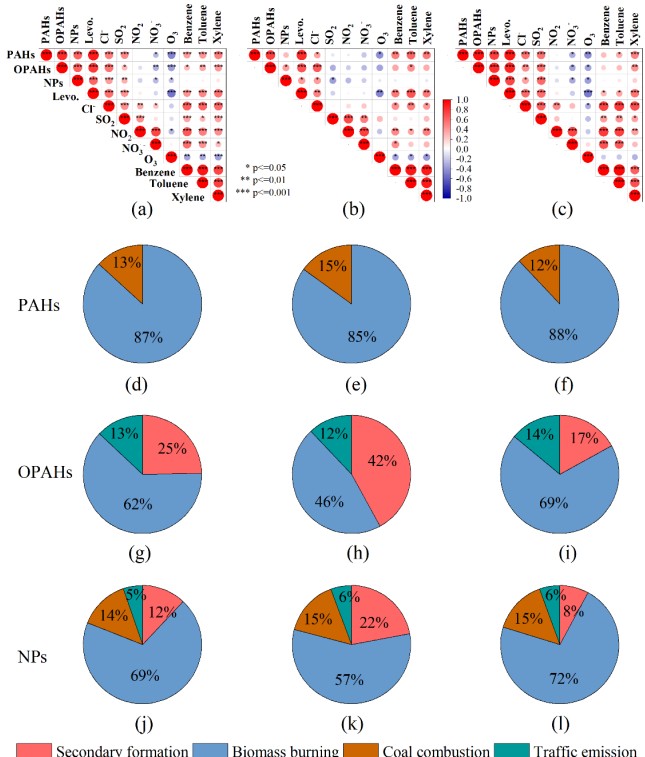

Fig.6 Correlations between PAHs, OPAHs, and NPs and key gas pollutants and aerosol components throughout the entire campaign (a), before heating (b), and during heating (c) (* $p<0.05$, ** $p<0.01$, *** $p<0.001$); Source apportionments for PAHs (d, e, f), OPAHs (g, h, i), and NPs (j, k, l) during the whole campaign, before heating, and during heating, respectively; SF: secondary formation; FF: fossil fuel combustion; BB: biomass burning; CC: coal combustion; TE: traffic emission).

**3.3.2 Source contribution**

To further analyze the quantitative and qualitative effects of primary emissions as well as secondary formation, this study identified four categories of distinct sources during the campaign using the PMF model, as shown in Fig. S1. The first source factor, biomass burning, was recognized by the highest levoglucosan concentrations and increased particulate levels of $K^+$ and $Cl^-$ within $PM_{2.5}$ (Fig. S1a). Biomass burning





emerged as the predominant source of aromatic compounds in the urban atmosphere of
Dongying during autumn and winter (Fig. 6d-l), contributing 87, 62, and 69% of the
total PAHs, OPAHs, and NPs, respectively, throughout the entire campaign (Fig.6d, g,
j). Moreover, the proportional contributions of burning biomass were also higher while
heating than before heating. A large number of PAHs, along with their derivatives,
originate from biomass combustion, as reported by various previously published studies
(Yhab et al., 2021; Chong et al., 2021; Bai et al., 2023; Li et al., 2022; Luo et al., 2021).
Several previous studies have also established that NPs are directly released during
biomass combustion and found emission factors over a range of 0.4 to 11.1 mg kg$^{-1}$
(Hoffmann et al., 2007; Iinuma et al., 2007; Wang et al.,2017b).

The second source factor was identified as coal combustion, having the highest

SO$_2$ concentration (Fig. S1b). The average contribution of coal combustion to PAHs
and NPs was determined as 13% and 14%, respectively (Fig.6d, j). Furthermore, there
were no significant changes to the relative contributions prior to and during the heating
phase. Previous research has indicated that coal combustion-related activities
significantly contribute to elevated levels of particulate PAHs and NPs, particularly
during the heating season. (Lu et al., 2019; Wang et al., 2018; Ren et al., 2023; Cai et
al., 2022; Luo et al., 2021).

In comparison to the other factor profiles, the third source factor, traffic emission,

showed a higher NO concentration loading in this factor profile (Fig. S1c). The average
contribution of traffic emissions to OPAHs throughout the entire campaign was 13%,
with no substantial variation in the percentage of emissions before and during heating



(14%). There was a minimal change in the contribution of traffic emissions to NPs both
before and during heating; on average, these emissions accounted for 6% of total NPs
during the campaign (Fig.6j-l). Previous studies have identified the traffic emission as
an essential source of OPAHs (Chong et al., 2021; Wang et al., 2022), and road traffic
as a primary contributor to NPs (Zhang et al., 2010; Ren et al., 2022). These differences
can be attributed to variations in energy use and industrial structures across different
cities.

Secondary formation, the fourth factor, displayed high concentrations of $O_3$ and a

relatively higher abundance of $SO_4^{2-}$, $NO_3^-$, and $NH_4^+$ (Fig. S1d). Secondary formation
contributed an average of 25% of OPAHs throughout the whole campaign (Fig.6g- i).
It was the second-largest source, with a significantly higher contribution before heating
(42%) compared to during heating (17%). However, secondary formation contributed
12% of NPs throughout the whole campaign, with higher contributions before heating
(22%) (Fig.6j- l). The vital role of secondary formation in contributing to both OPAHs
and NPs agreed with recent studies (Ren et al., 2024; Ren et al., 2022). Previous
modeling and field studies also identified secondary formation as a significant NPs
source in the atmosphere (Yuan et al., 2016b; Mayorga et al., 2021; Xie et al., 2017).
**3.4 Secondary formation mechanism based on quantum chemical calculations**

Aromatic hydrocarbons are a major class of VOCs that play a vital role in the

production of secondary organic aerosols (SOA), particularly in urban areas (Song et
al., 2021). However, this study reported the positive correlation of these aromatic





compounds with the key precursors (i.e. benzene, toluene, xylene), while a negative
correlation with $NO_2$, $NO_3^-$ and $O_3$, which are key parameters during the reaction
process (Fig.6a-c). Therefore, these substances were primarily derived from the same
source as the precursors, rather than their main products. Furthermore, considering the
substantial effect of biomass combustion, these compounds are probably the secondary
products formed through the oxidation process of gaseous pollutants released during
the biomass burning process. Nitroaromatic hydrocarbons are critical species in the
environment that are influenced by biomass combustion emissions. Phenol,
nitrobenzene, and *P*-Cresol are their significant precursors (Wang et al., 2020).
Therefore, to better comprehend the formation pathways of NPs, density functional
theory calculations at the M06-2X/6-311++G(2df,2p) level of theory were carried out
to explore the oxidation processes of major precursors by $NO_2$ and OH, specifically for
the dominant species 4NP and 4M5NC (Huang et al., 2010; Roman et al., 2022;
Shenghur et al., 2014; Wang et al., 2019b). Electronic energies (EDFT), Gibbs free
energies (GDFT) as well as the imaginary frequency of the reactant monomers, reactant
complexes (RC), transition states (TS), intermediate (IM) and product complexes (PC)
at 298.15 K and 1 atm were collected in Tables S1-S4 of the Supplementary Information.
The overall formation mechanisms, detailed in Fig. 7a, include both H-abstraction and
OH/$NO_2$ addition steps where Mechanisms 1 and 2 show the two-step formation
pathways for 4NP, while Mechanism 3 describes the four-step pathway for 4M5NC.
In the H-abstraction step, either an OH radical or $NO_2$ molecule abstracts a
hydrogen atom from the aromatic ring, forming an intermediate free radical (IM) and





resulting in the formation of $H_2O$ or HONO. In the $OH/NO_2$ addition step, the IMs react
with OH or $NO_2$ without any reaction barriers. In Table S1 of the Supplementary
Material, the calculated binding energy ($E$), Gibbs free energy change ($\Delta G$) for the
reactant monomers, Gibbs free energy ($G$), transition states (TS), reactant complexes
(RC), intermediate (IM) and PCs are presented. While Fig.7b exhibited the calculated
dominant formation mechanism of 4NP and 4M5NC along with the reaction barrier in
kcal/mol.
For the formation of 4NP, two precursors, phenol and nitrobenzene, were selected
which contain three distinct types of hydrogen atoms in their aromatic rings (*ortho*-,
*meta*-, and *para*-H). Here only the *para*-H is considered for the formation of 4NP.
During the daytime, when the gas-phase concentration of OH radicals is significant,
phenol initially reacts with OH to yield a reactant complex $RC_{ph-OH}$ with a $\Delta G$ of 3.16
kcal/mol. Then $RC_{ph}$ must overcome a reaction barrier of 8.64 kcal/mol to transition to
the $TS_{ph-OH}$ state, resulting in the generation of IM1, a free radical intermediate with a
$\Delta G$ of -1.03 kcal/mol. The overall H-abstraction step was slightly exothermic, with a
reaction barrier of 11.80 kcal/mol. At nighttime, with a sharp decrease in OH
concentrations and a relatively higher level of $NO_2$, the daytime-generated IM1 rapidly
proceeded through a reaction with $NO_2$ that has no energy barrier, resulting in the
forming 4NP with a $\Delta G$ of -64.50 kcal/mol.
During nighttime, a parallel H-abstraction step occurred where $NO_2$, instead of
OH, abstracted a hydrogen atom from phenol. However, this step had high overall
reaction barriers, reaching 40.31 kcal/mol in the gas phase and 40.54 kcal/mol in the

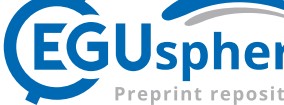

liquid phase (water). Moreover, the reaction was highly endothermic with the $\Delta G$ of
36.51 and 36.45 kcal/mol in the gas phase and liquid phase, respectively, indicating that
the H-abstraction step initiated by $NO_2$ was not feasible. Therefore, the calculations for
Mechanisms 2 and 3 focused solely on the H-abstraction step initiated by OH.

For an alternative formation pathway of 4NP from nitrobenzene during the

daytime, the overall H-abstraction step was slightly exothermic (-1.08 kcal/mol) with a
reaction barrier of 12.66 kcal/mol. The generated IM2 then proceeded through a highly
exothermic and barrier-free OH addition step to form 4NP. In a comparative analysis,
it was possible to initiate 4NP formation with both phenol and nitrobenzene. For
mechanism 1, phenol was preferred to undergo the H-abstraction reaction during the
daytime and the $NO_2$ addition reaction at nighttime. For mechanism 2, nitrobenzene
preferred to undergo both reactions involving H-abstraction and OH addition during the
daytime.




Fig.7 (a) Suggested formation mechanism for 4NP and 4M5NC (b) The proposed

dominant formation mechanism of 4NP and 4M5NC at the M06-2X/6-

311++G(2df,2p) level of theory. H, C, S, N, and O atoms are depicted as white, gray,

yellow, blue, and red spheres, respectively. The reaction barrier presented in the

parentheses are given in kcal/mol.

The formation of 4M5NC followed a four-step mechanism with *p*-Cresol as the

precursor. During the daytime, *p*-Cresol initially reacted with OH to form free radical

intermediate IM3, overcoming a reaction barrier of 7.19 kcal/mol. After that, IM3

underwent a highly exothermic OH addition step without any energy barrier to form

IM4 which was then proceeded through a reaction barrier of 11.83 kcal/mol to produce

IM5. At nighttime, the IM5 formed during the daytime quickly reacted through a highly

exothermic $NO_2$ addition step with no significant energy barrier, resulting in the





formation of 4M5NC, with a ΔG of -61.14 kcal/mol in the gas phase and -63.47
kcal/mol in the liquid phase.
**4 Conclusions**
PAHs, OPAH, and NACs in the $PM_{2.5}$ samples were analyzed during autumn and
winter in Dongying, a petrochemical industrial city. The concentrations of all these
species were significantly greater during the heating period in comparison to before
heating. Furthermore, these compounds demonstrated higher levels at night than during
the day, whereas NPs showed no notable diurnal variation. BbF, 1-NapA, and 4M5NC
were found to be the most abundant species for PAHs, OPAH, and NACs, respectively.
Biomass burning emerged as the primary source of aromatic compounds, particularly
during the heating period. Quantum chemical calculations revealed that the formation
mechanisms of 4NP and 4M5NC involve both H-abstraction and $OH/NO_2$ addition
steps with the H-abstraction step serving as the rate-limiting step, while the $OH/NO_2$
addition step proceeded without an energy barrier. Phenol and *P*-Cresol were
determined as the primary precursors for the formation of 4NP and 4M5NC,
respectively. This study offers the first detailed investigation into the sources and
forming mechanisms of aromatic compounds in the atmosphere of petrochemical cities
in the Yellow River Delta. It offers essential data and strategic guidance for reducing
aromatic compound emissions in such urban environments.



**Data availability**

The Data used in this study are available from the first author upon request (Yanqin Ren via renyq@craes.org.cn).

**Author contributions**

YR, GW and YJ designed the research; YJ, FB and ZW collected the samples and analyzed the data; YR and HZ wrote the manuscript; JL, RG, FL, ZL and HL contributed to the paper with useful scientific discussions and comments.

**Competing interests**

The contact author has declared that none of the authors has any competing interests.

**Acknowledgements**

We are grateful to Mr. Xiaoyu Yan and Ms. Xurong Bai from Chinese Research Academy of Environmental Sciences for their help in collecting samples, and Mr. Yubao Chen and his colleagues from East China Normal University for providing help with analytical testing.



**Financial support**

This research was supported by the National Research Program for Key Issues in

Air Pollution Control (No. DQGG202121, DQGG2021301), Key Technologies
Research and Development Program (No. 2023YFC3706105), Fundamental Research
Funds for Central Public Welfare Scientific Research Institutes of China (No.
2019YSKY-018), and the National Natural Science Foundation of China (No.
42130704; No. 41907197; No.22306179).

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
