# Peer review of "Significant contributions of biomass burning to PM2.5-bound aromatic"

_EGUsphere, 2024_

## Referee Comment (RC1)

**Referee Comments – Significant contributions of biomass burning to PM$_{2.5}$-bound aromatic compounds: insights from field observations and quantum chemical calculations (https://doi.org/10.5194/egusphere-2024-3678)**

**General Overview:**

The manuscript (egusphere-2024-3678) presents an interesting study investigating the sources and formation mechanisms of aromatic compounds, specifically focusing on polycyclic aromatic compounds (PAHs), oxygenated PAHs (OPAHs), and nitrated phenols (NPs) in the atmosphere of Dongying, a city in the Yellow River Delta. The topic of this study falls within the scope of the journal Atmospheric Chemistry and Physics (ACP). The authors present data from field observations and analyze the data with a source apportionment model and quantum chemical calculations. The authors provide insights and helpful information for reducing emissions of aromatic compounds in Dongying and other places with similar atmospheric environments. This manuscript is generally laid out well and shows its academic value. This manuscript is recommended to be published after addressing the concerns and comments below with minor revisions.

**Major Concern:**

- Lines 277 – 281: The 2 sentences seem to contradict each other. The first sentence says that there is not significant diurnal variation, while the second sentence says that there is a significant nighttime increase compared with daytime. It would be great if these 2 sentences could be explained in more detail or modified to make sure that the statements throughout this manuscript are consistent with each other.

**Minor Concerns:**

- Line 30: What is heating season? Does it mean winter or from which months to which months? It would be great to be clear in the description for the readers to avoid confusion.

- Lines 32 – 33: Is the term "positive matrix factorization" the full form of the acronym PMF? If so, please clarify this in the text for readers to know what PMF stands for later in the text.

- Line 34: Is the term "density functional theory" the full form of the acronym DFT? If so, please clarify this in the text for readers to know what DFT stands for later in the text.

- Line 39: What is the definition of a petrochemical city in this study? What specifically are the cities being referred to in this sentence? It seems that the city Donying is the only focus throughout this study.

- Line 104: It is strongly encouraged that a map with the sampling site and the land use information of surrounding areas labeled is provided.

- Line 118: How was this quarter of filter obtained and quantified to make sure it is actually a quarter (25%)? If it is based on the surface area of the filter, then the weight of sample collected on each quarter of the filter might be different and could be confirmed by weighing to quantify the amount of sample, which might deviate from the presumed 25%. The accuracy of the analysis should be enhanced by taking this deviation into account. No matter if this step of weighing each quarter of filter is taken or not, the description should be clarified in the manuscript.

- Lines 133 – 150: It would be more reader-friendly if those kinds of 13 different PAHs, 8 different OPAHs, 9 different NPs, and 8 water-soluble inorganic ions are summarized as a list in a table instead of putting them in a paragraph. Readability of an article helps readers find useful information easily and thus potentially increases the opportunity of citations for this article in the future.

- Line 169: Is the quartz filter a "fiber" filter or a "membrane" filter? Please double check.

- Lines 169 – 170: What is the purpose of this pretreatment procedure for the filters? Is it trying to remove adsorbed contaminants on the surfaces of quartz fiber filters? If so, it would be great to mention that for clarity.

- Lines 119 and 172: If it is a fiber filter rather than a membrane filter, the word "membrane" is suggested to be removed.

- Line 184: Starting from Line 184 moving forward, do the numbers represent mean plus or minus one standard deviation or two standard deviations or something else? It would be great if they are mentioned explicitly.

- Lines 189 – 190: Is there any information or data supporting this statement of simply attributing all the increases of measured gaseous air pollutants to the increased burning activities? Are there any relevant data showing that other emission sources of the pollutants remain unchanged or unimportant during the study periods? Could

other factors such as meteorology be attributable to such increases as well? Since there are many factors influencing the trends of pollutants, the discussion here is suggested to be expanded. If the authors believe that the discussion is already covered in a later section of this article, please also mention that explicitly to avoid confusion.

- Lines 193 – 194: Is there solar radiation information to support this statement of ascribing the variation of gaseous pollutants before heating?

- Lines 195 – 196: There are variations in $PM_{2.5}$ concentration throughout the sampling period according to Figure 1(d). It might not be appropriate to oversimplify the analysis of the trend by simply looking at the average value over a period of several days and by saying that the concentrations remain unchanged, especially when the data of time series are available.

- Line 201: Why is 1.6 used to convert OC to OM? Are there studies in the literature to refer to? If so, please add citations.

- Line 204: If the average values are to be mentioned, Table 1 rather than Fig. 1(b) and (c) should be referred to. If Fig. 1 is to be referred to, it should be Fig. 1(d) so that there are data of the species mentioned.

- Lines 204 – 205: The total concentrations and proportions of secondary inorganic aerosols are both mentioned initially, but only the average of proportion (35%) is mentioned in the following statement of the same sentence. It would be great if the information of concentration is also mentioned.

- Lines 211 – 213: Are there any studies in the literature to support the statement that OC is generated from both primary emissions and secondary formation, while EC is predominantly derived from primary sources? If so, please add citations.

- Lines 213 – 215: Are there any studies in the literature to support the statement that the significant increase in the concentration of OC alongside a stable OC/EC ratio suggests that the sources of carbonaceous aerosols in this study include both primary emissions and secondary formation processes? If so, please add citations.

- Lines 215 – 219: Is r = 3 for potassium ion ($K^+$) considered as high? Are there any studies in the literature to support the statement? If so, please add corresponding citations.

- Table 1 and Table 2: Does N mean the number of samples? It would be great if the definition of N for the tables could be explicitly mentioned. Similar to the comment on Line 184, do the numbers for temperature, relative humidity, gaseous pollutants,

PAHs, etc. represent mean plus or minus one standard deviation or two standard deviations or something else? It would be great if they are mentioned explicitly. For the numbers in the parentheses, do they mean minimum to maximum or $5^{th}$ percentile to $95^{th}$ percentile? It would be great if they are explicitly mentioned. What does NA mean? Does it mean lower than the limit of detection or the limit of quantification? If so, what are the corresponding limit of detection and limit of quantification?

- Line 223: What are SOR and NOR in Fig. 1(c)? Please define them.

- Line 230: Please define r, p, and n in Fig. 3. Is simple linear regression or any other type of regression analysis used here? Please clarify.

- Lines 246 – 247: The pattern aligns with observations from other locations in previous studies. Is there any analysis or explanation for the results? Is this pattern due to the diel variations of mixing heights or other factors?

- Lines 249 – 250: The dominance of BbF was found to be consistent with the previous research. Is there any reason to potentially explain this?

- Lines 253 – 254: The average during the heating period is higher than that before the heating period, but the maximum value before the heating period is higher than that during the heating period. Is there any potential explanation for the higher maximum before heating?

- Lines 255 – 257: Is this observation due to mixing height or other factors?

- Lines 260 – 264: Is there any explanation for such difference compared with other cities?

- Lines 274 – 275: What is the relation between Before heating, During heating, autumn, and winter? Some clarifications might be needed to enhance the readability.

- Line 277: To help clarify the definitions of terms used, it might be helpful if the dates defined for autumn, winter, summer, and spring are mentioned so that readers can more easily follow the flow of the description and compare with the "before heating" and "during heating" in this study, which have been defined clearly.

- Lines 281 – 283: Are there any data supporting the statement that the relative nine NPs molecular composition in $PM_{2.5}$ remained consistent throughout the observation period? It does not seem to be delivered in Fig. 4 or Fig. 5.

- Lines 283 – 286: The relative percentage is hard for the readers to observe by simply looking at Fig. 4 and Fig. 5 because the data shown are in concentrations rather than

percentages. It would be great if the figures showing relative percentages could be provided, either in the supplementary information document or the main article, for the discussion.

- Lines 286 – 294: Is there any explanation for the similarities and differences while comparing with the results from other cities?

- Line 298: In the Fig. 4, it shows "Before heating" and "During heating" while the caption shows autumn and winter. What are the definitions of autumn and winter here? Please be consistent with the terminology or clearly state the relation between different terms to avoid confusion.

- Line 303: If the whiskers in Fig. 5(a), (b), and (c) denote the 25$^{th}$ and 75$^{th}$ percentiles, what are the boxes for the box-whisker plots? Usually, the boxes show the 25$^{th}$ and 75$^{th}$ percentiles as well as the median values while the whiskers show either the 5$^{th}$ and 95$^{th}$ percentiles or minimum and maximum values. Please double check the values used in the Fig. 5. Additionally, what do the error bars indicate in Fig. 5(d), (e), and (f)? Do they indicate one standard deviation or 75$^{th}$ percentile or 95$^{th}$ percentile or maximum?

- Fig. 6: The numbers and text in Fig. 6(a), (b), and (c) are hard to read due to the small size. Fig. 6(a), (b), and (c) are suggested to be enlarged for readability.

- Line 334: Does "PMF" mean positive matrix factorization? It should be defined first before the acronym is used.

- Line 338: Does autumn mean before heating and winter mean during heating? Please clearly describe the relation between different terms or use the same terminology throughout the article to be consistent.

- Lines 369 – 372: Is there any explanation for the reason why the contribution from secondary formation is higher before heating?

- Lines 393 – 394: Should the DFT be a subscript?

- Lines 403 – 404: Does supplementary material mean the same thing as supplementary information mentioned in Line 396? If so, please use consistent terminology to avoid confusion.

- Line 411: Is there any reason to consider only the *para*-H for the formation of 4NP?

- Lines 440 – 441: Is there any S atom involved in the mechanisms?

**Technical Comments:**

- Line 46: The phrase "by the presence benzene ring" seems to be intended to mean "by the presence of benzene ring" but missed the word "of".

- Line 156: The word "theotry" seems to be a typographical error of "theory".

- Line 201: The word "component" seems to be a typographical error of "components" since the plural form should be used.

- Line 249: The word "accounting" seems to be a typographical error of the phrase "accounting for".

- Line 310: The word "substantial" seems to be a typographical error of "substantially".

- Lines 453 and 458: The word "OPAH" seems to be a typographical error of "OPAHs".

- Line 455: The phrase "in comparison to before heating" seems to be intended to mean "in comparison to those before heating" but missed the word "those".

---

## Author Response (AR1)

**Responses to the comments from reviewer #1**

**General Overview:**

The manuscript (egusphere-2024-3678) presents an interesting study investigating the sources and formation mechanisms of aromatic compounds, specifically focusing on polycyclic aromatic compounds (PAHs), oxygenated PAHs (OPAHs), and nitrated phenols (NPs) in the atmosphere of Dongying, a city in the Yellow River Delta. The topic of this study falls within the scope of the journal Atmospheric Chemistry and Physics (ACP). The authors present data from field observations and analyze the data with a source apportionment model and quantum chemical calculations. The authors provide insights and helpful information for reducing emissions of aromatic compounds in Dongying and other places with similar atmospheric environments. This manuscript is generally laid out well and shows its academic value. This manuscript is recommended to be published after addressing the concerns and comments below with minor revisions.

**Major Concern:**

1. Lines 277 – 281: The 2 sentences seem to contradict each other. The first sentence says that there is not significant diurnal variation, while the second sentence says that there is a significant nighttime increase compared with daytime. It would be great if these 2 sentences could be explained in more detail or modified to make sure that the statements throughout this manuscript are consistent with each other.

**Response:** Suggestion taken. The authors have modified these two sentences, and the relevant descriptions are as follows: "Different from OPAHs and PAHs, the total concentrations of ∑9NPs did not demonstrate significant diurnal variation before heating, but a significant nighttime increase was observed during the heating period…". **See lines 305-308**.

**Minor Concerns:**

2. Line 30: What is heating season? Does it mean winter or from which months to which months? It would be great to be clear in the description for the readers to avoid confusion.

**Response:** Suggestion taken. The authors have added a date description for "heating season" (line 30), and there are also specific descriptions of "before heating" and "during heating" in the main body, **see lines 187-190, Table 1**.

3. Lines 32 – 33: Is the term "positive matrix factorization" the full form of the acronym PMF? If so, please clarify this in the text for readers to know what PMF stands for later in the text.

**Response:** Suggestion taken. The authors have added the abbreviation "PMF" after "positive matrix factorization", **see line 33**.

4. Line 34: Is the term "density functional theory" the full form of the acronym DFT? If so, please clarify this in the text for readers to know what DFT stands for later in the text.

**Response:** Suggestion taken. The authors have added the abbreviation "DFT" after "density functional theory", **see line 35**.

5. Line 39: What is the definition of a petrochemical city in this study? What specifically are the cities being referred to in this sentence? It seems that the city Donying is the only focus throughout this study.

**Response:** In this study, we define a "petrochemical city" as an urban area where petrochemical industries constitute the dominant economic sector and primary emission source of aromatic compounds, characterized by concentrated petrochemical infrastructure (e.g., refineries, chemical plants, storage facilities) and associated emission activities. While our methodological framework is applicable to similar cities in the Yellow River Delta region, we acknowledge that the current investigation specifically focuses on Dongying City as a representative case study of petrochemical cities. Therefore, the authors modified the relevant sentences to avoid similar misunderstandings as follows: "…petrochemical cities, using Dongying in the Yellow River Delta as a representative case study. This investigation…". **See lines 39-41**.

6. Line 104: It is strongly encouraged that a map with the sampling site and the land use information of surrounding areas labeled is provided.

**Response:** Suggestion taken. The authors have added a map of the sampling site in the revised manuscript, including the main functional areas around it. **See Fig. 1**.

7. Line 118: How was this quarter of filter obtained and quantified to make sure it is actually a quarter (25%)? If it is based on the surface area of the filter, then the weight of sample collected on each quarter of the filter might be different and could be confirmed by weighing to quantify the amount of sample, which might deviate from the presumed 25%. The accuracy of the analysis should be enhanced by taking this deviation into account. No matter if this step of weighing each quarter of filter is taken or not, the description should be clarified in the manuscript.

**Response:** Thank you for your thorough review and valuable feedback. In this study, the filter was physically divided into four equal sectors by surface area and one quarter was used for analysis. While this approach is widely adopted in the field for sample subdivision (Liu, X., et al., 2023; Song, J., et al., 2022; Li, J., et al., 2020), we acknowledge your concern that uneven particle distribution across the filter could lead to deviations between the actual sample mass and the theoretical 25% value. In subsequent studies, we will integrate gravimetric validation of filter segments to account for mass distribution variability. As requested, the methodology section has been revised to clarify the procedure as follows: "…the sampled filter was divided into four equal surface area segments, and one quarter was utilized for experimental analysis. The quarter of the sampled filter…". **See lines 122-124**.

**References:**

Liu, X., et al., Secondary Formation of Atmospheric Brown Carbon in China Haze: Implication for an Enhancing Role of Ammonia. Environmental Science & Technology, 2023. 57: 11163-11172.

Song, J., M. Li, and C. Zou, Molecular characterization of nitrogen-containing compounds in humic-like substances emitted from biomass burning and coal combustion. Environmental Science & Technology, 2022. 56(1): 119-130.

Li, J., et al., Optical properties and molecular compositions of water-soluble and water-insoluble brown carbon (BrC) aerosols in Northwest China. Atmospheric Chemistry and Physics, 2020. 20: 4889–4904.

8. Lines 133 – 150: It would be more reader-friendly if those kinds of 13 different PAHs, 8 different OPAHs, 9 different NPs, and 8 water-soluble inorganic ions are summarized as a list in a table instead of putting them in a paragraph. Readability of an article helps readers find useful information easily and thus potentially increases the opportunity of citations for this article in the future.

**Response:** Based on all reviewer suggestions, specific species of PAHs, OPAHs, and NPs were listed in **Table S1**. And the authors also give the chemical structures, the vapor pressures and LOD of these species. **See lines 140-143, and Table S1**.

9. Line 169: Is the quartz filter a "fiber" filter or a "membrane" filter? Please double check.

**Response:** The authors changed the description and unified it into "quartz microfiber filter". **See lines 115, 171**.

10. Lines 169 – 170: What is the purpose of this pretreatment procedure for the filters? Is it trying to remove adsorbed contaminants on the surfaces of quartz fiber filters? If so, it would be great to mention that for clarity.

**Response:** Suggestion taken. The authors have added the related description, i.e. "…to remove any organic contaminants". **See line 172**.

11. Lines 119 and 172: If it is a fiber filter rather than a membrane filter, the word "membrane" is suggested to be removed.

**Response:** The word "membrane" was removed. **See lines 122, 174**.

12. Line 184: Starting from Line 184 moving forward, do the numbers represent mean plus or minus one standard deviation or two standard deviations or something else? It would be great if they are mentioned explicitly.

**Response:** Based on all reviewer suggestions, the numbers are revised into the average values. **See line 194, Tables1 and 2**.

13. Lines 189 – 190: Is there any information or data supporting this statement of simply attributing all the increases of measured gaseous air pollutants to the increased burning activities? Are there any relevant data showing that other emission sources of

the pollutants remain unchanged or unimportant during the study periods? Could other factors such as meteorology be attributable to such increases as well? Since there are many factors influencing the trends of pollutants, the discussion here is suggested to be expanded. If the authors believe that the discussion is already covered in a later section of this article, please also mention that explicitly to avoid confusion.

**Response:** Thank you for your careful review and constructive comments on the key conclusions. As the reviewer said, $SO_2$ and NO concentration variations are indeed potentially influenced by a variety of factors, such as meteorological conditions. However, no policies related to major industrial restructuring or traffic control policy changes were issued during the study period, and subsequent studies in this manuscript also show a strong positive correlation between $SO_2$ and burning markers during heating (e.g. PAHs, OPAHs and NPs) (r>0.55, $p$<0.01, **Sec. 3.3, Fig. 7c**). So here the authors attribute it mainly to changes in combustion sources. In order to make the expression much clearer and avoid arbitrary, the authors modified the sentence as follows: "These increases are primarily associated with enhanced combustion activities during the heating season, which supported by the correlation analysis (**Sec. 3.3**) though synergistic effects (e.g., meteorological effect) require further quantification in future studies". **See lines 198-201**.

14. Lines 193 – 194: Is there solar radiation information to support this statement of ascribing the variation of gaseous pollutants before heating?

**Response:** Thank you for raising this important point. We acknowledge that direct solar radiation measurements were not available in our study. The reference to "stronger solar radiation" was based on the well-established positive correlation between elevated temperatures and enhanced light conditions (e.g., sunlight intensity and duration). To improve clarity and rigor, we have revised the relevant statement in the revised manuscript to: "This variation was associated with the higher temperatures and light conditions before heating, which are conductive to photochemical reactions". **See lines 204-206**.

15. Lines 195 – 196: There are variations in PM2.5 concentration throughout the sampling period according to Figure 1(d). It might not be appropriate to oversimplify the analysis of the trend by simply looking at the average value over a period of several days and by saying that the concentrations remain unchanged, especially when the data of time series are available.

**Response:** Thanks to the reviewer's suggestion, the authors admit that the conclusion of this sentence is somewhat arbitrary. And considering that the paragraph in this sentence mainly describes the change of the main components of $PM_{2.5}$, the authors have deleted this sentence in the revised manuscript.

16. Line 201: Why is 1.6 used to convert OC to OM? Are there studies in the literature to refer to? If so, please add citations.

**Response:** Suggestion taken. The authors have added the citation (**line 211**) as follows:

Turpin, B. J. and Lim, H. J.: Species Contributions to PM$_{2.5}$ Mass Concentrations: Revisiting Common Assumptions for Estimating Organic Mass, Aerosol Science & Technology, 35:1, 602-610, https://doi.org/10.1080/02786820119445, 2001.

17. Line 204: If the average values are to be mentioned, Table 1 rather than Fig. 1(b) and (c) should be referred to. If Fig. 1 is to be referred to, it should be Fig. 1(d) so that there are data of the species mentioned.
**Response:** Sorry for this mistake, the authors have changed it to **Table 1. See line 214**.

18. Lines 204 – 205: The total concentrations and proportions of secondary inorganic aerosols are both mentioned initially, but only the average of proportion (35%) is mentioned in the following statement of the same sentence. It would be great if the information of concentration is also mentioned.
**Response:** Suggestion taken. The authors have added the concentration value. **See lines 215-216**.

19. Lines 211 – 213: Are there any studies in the literature to support the statement that OC is generated from both primary emissions and secondary formation, while EC is predominantly derived from primary sources? If so, please add citations.
**Response:** Suggestion taken. The authors have added the citation (**line 223**) as follows:
Zhang, R., Wang, G., Guo, S., L. Zamora, M., Ying, Q., Lin, Y., Wang, W., Hu, M., and Wang, Y.: Formation of Urban Fine Particulate Matter, Chem. Rev., 115, 3803-3855, 2015.

20. Lines 213 – 215: Are there any studies in the literature to support the statement that the significant increase in the concentration of OC alongside a stable OC/EC ratio suggests that the sources of carbonaceous aerosols in this study include both primary emissions and secondary formation processes? If so, please add citations.
**Response:** Suggestion taken. The authors have added the citation (**line 226**) as follows:
Zhang, R., Wang, G., Guo, S., L. Zamora, M., Ying, Q., Lin, Y., Wang, W., Hu, M., and Wang, Y.: Formation of Urban Fine Particulate Matter, Chem. Rev., 115, 3803-3855, 2015.

21. Lines 215 – 219: Is r = 0.3 for potassium ion (K$^+$) considered as high? Are there any studies in the literature to support the statement? If so, please add corresponding citations.
**Response:** Upon re-examining the data, we realize there was an error in our original description of the correlation coefficients. And we acknowledge that the correlation coefficient (r = 0.3) between levoglucosan and K$^+$ during heating is indeed relatively moderate. We have revised the text to reflect these nuances clearly as follows:
A strong positive correlation between levoglucosan and Cl$^-$ (r =0.58–0.69) and a weaker correlation with K$^+$ (r = 0.3–0.34) were observed (**Fig. 4**), with both ions

showing elevated concentrations during the heating period, suggests additional combustion activities, e.g. biomass burning. **See lines 226-230**.

22. Table 1 and Table 2: Does N mean the number of samples? It would be great if the definition of N for the tables could be explicitly mentioned. Similar to the comment on Line 184, do the numbers for temperature, relative humidity, gaseous pollutants, PAHs, etc. represent mean plus or minus one standard deviation or two standard deviations or something else? It would be great if they are mentioned explicitly. For the numbers in the parentheses, do they mean minimum to maximum or 5th percentile to 95th percentile? It would be great if they are explicitly mentioned. What does NA mean? Does it mean lower than the limit of detection or the limit of quantification? If so, what are the corresponding limit of detection and limit of quantification?

**Response:** Suggestion taken. The authors have added the data description (i.e. Mean (Min-Max)) and comments at the bottom of the tables. As follows: [a] *N*: the number of samples, and [b] *NA*: not available. **See Tables1 and 2**.

23. Line 223: What are SOR and NOR in Fig. 1(c)? Please define them.

**Response:** Considering the comments of all reviewers, the authors have moved it from **Fig. 2 in the revised manuscript**.

24. Line 230: Please define r, p, and n in Fig. 3. Is simple linear regression or any other type of regression analysis used here? Please clarify.

**Response:** The authors have modified the title of **Fig. 4** and provided explanations for the relevant parameters. **See lines 241-246**.

25. Lines 246 – 247: The pattern aligns with observations from other locations in previous studies. Is there any analysis or explanation for the results? Is this pattern due to the diel variations of mixing heights or other factors?

**Response:** The authors have added the description as follows: "…and this diurnal variation is strongly influenced by the emission sources (e.g. coal and biomass burning, heavy truck traffic), in addition to air temperature and boundary layer height". **See lines 261-264**.

26. Lines 249 – 250: The dominance of BbF was found to be consistent with the previous research. Is there any reason to potentially explain this?

**Response:** The authors have added the description as follows: "This molecular composition mainly affected by the emission source, and low efficiency combustion activities emitting more BbF are still very common in China (Wang et al., 2009)". **See lines 267-269**.

**Reference:**

Wang, G., et al., Size-distributions of n-alkanes, PAHs and hopanes and their sources in the urban, mountain and marine atmospheres over East Asia. Atmospheric Chemistry & Physics, 2009. 9(22): 8869-8882.

27. Lines 253 – 254: The average during the heating period is higher than that before the heating period, but the maximum value before the heating period is higher than that during the heating period. Is there any potential explanation for the higher maximum before heating?

**Response:** Thank you very much for pointing out this very important finding. According to the reviewer's suggestion, the authors have added the relevant description. As follows: "There is a peak concentration of OPAHs on Nov. 4, and it could be attributed to significant biomass burning activities during the night of Nov. 3, which likely enhanced both primary emissions and secondary formation processes of these compounds". **See lines 274-277**.

28. Lines 255 – 257: Is this observation due to mixing height or other factors?

**Response:** The authors have added the reason as follows: "Like other pollutants, this diurnal variation is influenced not only by the boundary layer height, but also by the emission source". **See lines 279-281**.

29. Lines 260 – 264: Is there any explanation for such difference compared with other cities?

**Response:** The authors have added possible reason as follows: "The significantly higher ∑8OPAHs concentrations in this study likely reflect localized emission sources, including mixed industrial-traffic-biomass burning activities, coupled with stagnant meteorological conditions during the sampling. Notably, stricter fuel quality standards and emission controls in the compared cities may also contribute to these disparities". **See line 290-294**.

30. Lines 274 – 275: What is the relation between Before heating, During heating, autumn, and winter? Some clarifications might be needed to enhance the readability.

**Response:** Suggestion taken. The authors have noted the months at the end of the season. As follows: "…during autumn and winter (18 Oct., 2017 – 17 Jan., 2018)…". **See line 302**.

31. Line 277: To help clarify the definitions of terms used, it might be helpful if the dates defined for autumn, winter, summer, and spring are mentioned so that readers can more easily follow the flow of the description and compare with the "before heating" and "during heating" in this study, which have been defined clearly.

**Response:** Similar to the previous question, the authors have noted the months at the end of the season. As follows: "…in the summer (1–30 July, 2017) (8.5 ng m$^{-3}$) and spring (10–21 April, 2017)…". **See lines 304-305**.

32. Lines 281 – 283: Are there any data supporting the statement that the relative nine NPs molecular composition in PM2.5 remained consistent throughout the observation period? It does not seem to be delivered in Fig. 4 or Fig. 5.

**Response:** Thank you for raising this important clarification. We appreciate the opportunity to refine our statement. The intended meaning is that the rank order of

relative abundance for the nine NP species remained unchanged during the observation period, not that their absolute molecular compositions were identical. This conclusion is supported by the data presented in **Fig. 5 and Table 2**. We have revised the text to better reflect this interpretation as follows: "…the rank order of relative abundance for the nine NP species in PM$_{2.5}$ remained consistent throughout the observation period." **See lines 309-311**.

33. Lines 283 – 286: The relative percentage is hard for the readers to observe by simply looking at Fig. 4 and Fig. 5 because the data shown are in concentrations rather than percentages. It would be great if the figures showing relative percentages could be provided, either in the supplementary information document or the main article, for the discussion.

**Response:** I'm sorry to cause such trouble to the reviewers. The relative proportions of each species are shown in **the pie chart in Fig. 5**. At the same time, the authors make a more explicit explanation after the relevant description. As follows: "…(the pie charts in Fig. 5d)". **See line 314**.

34. Lines 286 – 294: Is there any explanation for the similarities and differences while comparing with the results from other cities?

**Response:** Suggestion taken. The authors have added a probable explanation as follows: "The spatial differences in the above concentration levels were influenced by a combination of various factors. Besides meteorological conditions, the influence of different emission sources was very important". **See lines 322-325**.

35. Line 298: In the Fig. 4, it shows "Before heating" and "During heating" while the caption shows autumn and winter. What are the definitions of autumn and winter here? Please be consistent with the terminology or clearly state the relation between different terms to avoid confusion.

**Response:** The authors have changed **the caption of Fig. 5** to "Time series of levoglucosan (a), PAHs(b), OPAHs (c), and NPs (d) before and during heating in Dongying".

36. Line 303: If the whiskers in Fig. 5(a), (b), and (c) denote the 25th and 75th percentiles, what are the boxes for the box-whisker plots? Usually, the boxes show the 25th and 75th percentiles as well as the median values while the whiskers show either the 5th and 95th percentiles or minimum and maximum values. Please double check the values used in the Fig. 5. Additionally, what do the error bars indicate in Fig. 5(d), (e), and (f)? Do they indicate one standard deviation or 75th percentile or 95th percentile or maximum?

**Response:** Thank you for your thoughtful comments. The authors have revised **the caption of Fig. 6 in the revised manuscript** and added a note, as follows:
"…The top and bottom of the vertical line for each box in (a-c) correspond to the 95th and 5th percentiles, respectively, and the top, middle, and bottom horizontal lines of the box mark the 75th, 50th, and 25th percentiles of the data range. The white dot in

each box represents the mean value. The error bars in (d-f) indicate the standard deviation". **See lines 333-337**.

37. Fig. 6: The numbers and text in Fig. 6(a), (b), and (c) are hard to read due to the small size. Fig. 6(a), (b), and (c) are suggested to be enlarged for readability.
**Response:** Suggestion taken. The authors have redrawn **Fig. 7 in the revised manuscript**.

38. Line 334: Does "PMF" mean positive matrix factorization? It should be defined first before the acronym is used.
**Response:** Suggestion taken. The authors have added "positive matrix factorization" before the abbreviation "PMF", **see line 369**.

39. Line 338: Does autumn mean before heating and winter mean during heating? Please clearly describe the relation between different terms or use the same terminology throughout the article to be consistent.
**Response:** Suggestion taken. The authors have changed "autumn and winter" to "the whole campaign" here. **See line 374**. At the same time, the authors have revised the full text for similar problems.

40. Lines 369 – 372: Is there any explanation for the reason why the contribution from secondary formation is higher before heating?
**Response:** Suggestion taken. The authors have added the probable reason as follows: "The contribution of secondary formation to OPAHs and NPs before heating was about twice that of during heating, mainly because the higher temperature and stronger atmospheric oxidation before heating (**Sec.3.1**) were conducive to the chemical reactions that generate these species". **See lines 417-420**.

41. Lines 393 – 394: Should the DFT be a subscript?
**Response:** Yes. "EDFT" and "GDFT" have been modified into "$E_{DFT}$" and "$G_{DFT}$", respectively. **See line 442**.

42. Lines 403 – 404: Does supplementary material mean the same thing as supplementary information mentioned in Line 396? If so, please use consistent terminology to avoid confusion.
**Response:** Suggestion taken. The authors have changed into "SI". **See line 445**.

43. Line 411: Is there any reason to consider only the para-H for the formation of 4NP?
**Response:** The authors have added the reasons as follows: "In the case where the -OH and -NO$_2$ functional groups in 4NP are at the para-position, hence only the *para*-H in phenol and nitrobenzene were chosen for the formation of 4NP". **See lines 459-461**.

44. Lines 440 – 441: Is there any S atom involved in the mechanisms?
**Response:** Sorry for the error, the authors have removed the words of "S" and

"yellow" here. **See lines 489-491**.

**Technical Comments:**

45. Line 46: The phrase "by the presence benzene ring" seems to be intended to mean "by the presence of benzene ring" but missed the word "of".
**Response:** Suggestion taken. The authors have added the word "of". **See line 47**.

46. Line 156: The word "theotry" seems to be a typographical error of "theory".
**Response:** Suggestion taken. It has been modified into "theory". **See line 157**.

47. Line 201: The word "component" seems to be a typographical error of "components" since the plural form should be used.
**Response:** Suggestion taken. It has been modified into "components". **See line 210**.

48. Line 249: The word "accounting" seems to be a typographical error of the phrase "accounting for".
**Response:** Suggestion taken. The authors have added the word "for". **See lines 264-265**.

49. Line 310: The word "substantial" seems to be a typographical error of "substantially".
**Response:** Suggestion taken. The authors have changed it into "substantially". **See line 344**.

50. Lines 453 and 458: The word "OPAH" seems to be a typographical error of "OPAHs".
**Response:** Suggestion taken. The authors have changed them into "OPAHs". **See lines 512, 517**.

51. Line 455: The phrase "in comparison to before heating" seems to be intended to mean "in comparison to those before heating" but missed the word "those".
**Response:** Suggestion taken. The authors have added the word "those". **See line 514**.

**Responses to the comments from reviewer #2**

**General Comments**

In this study by Ren et al., the authors report measurements of polycyclic aromatic hydrocarbons (PAHs), oxidized PAHs (OPAHs) and nitrophenols (NP) at an industrial city in China. By comparing the concentration before and during the heating period and using PMF analysis, the authors conclude that major sources of these compounds at this location are biomass burning and coal combustion, with a minor contribution from secondary formation. The authors also perform quantum chemical calculations to identify the major reaction pathways for the formation of 4-nitrophenol and 4-methyl-5-nitrocresol. The manuscript is clear and well written, and within the scope of the journal. Some minor revisions are included below.

**Methods:**

1. It is never actually stated how the PAHs, OPAHs and NPs were analyzed.

**Response:** The authors have added a description of the analysis as follows: "The derivatized samples were examined using gas chromatography coupled with a mass spectroscopy detector (GC/MS: HP 7890A, HP 5975C, Agilent Co., USA). The extraction and derivatization methods described above allowed for the simultaneous measurement of the samples' polar and non-polar constituents. This study analyzed thirteen different PAHs, eight different OPAHs, and nine different NPs, as well as some organic tracers, e.g. levoglucosan. Specific species of PAHs, OPAHs, and NPs were listed in TableS1 in the Supplementary Information (SI)". **See line 136-143**.

2. Details are also needed about how PMF was run and why the four-factor solution was selected.

**Response:** Suggestion taken. Given the length of the text, the authors have included a detailed introduction to the PMF process **in the SI (Text S1)** and gave hints in the text, including **Table S2 and Fig. S1. See lines 369-371**. The PMF operation process in the SI as follows:

**Text S1: PMF operation process:**

To further analyze the quantitative and qualitative effects of primary emissions as well as secondary formation, we applied the Positive Matrix Factorization (PMF) (EPA PMF 5.0 version) receptor model. Daily measurements of $SO_2$, NO, $O_3$, $PM_{2.5}$, OC, EC, $SO_4^{2-}$, $NO_3^-$, $NH_4^+$, $Na^+$, $Cl^-$, $K^+$, levoglucosan (Lev.), PAHs, OPAHs, and NPs were used as input data. According to the results of the simulation, the categories of EC and $Na^+$ were found to be bad. Numerous runs of the model using three to six factors and different combinations of the absorption and concentration data set were carried out. The agreement between the model fit and the correlation between estimated and measured concentrations are respectively depicted by the values of Q

and $r^2$, and the choice of proper factor number for modeling relies on these values (Comero et al., 2009). **Table S2** gives the outcome of Q robust, average $r^2$, and Q true value. With the dQ value under 5% of the Base Run Q (robust), the Fpeak results were acceptable. Moreover, there is a reasonable physical interpretation of all factors via the evaluation of the variation in maximum individual column standard deviation parameters and the maximum individual column mean since they demonstrated a notable drop with the change in the number of factors in this study (Cesari et al., 2017). All these results indicated the number of factors (four factors) was appropriate (**Fig. S1**).

**References**

Cesari, D., Benedetto, G. E. D., Bonasoni, P., Busetto, M., and Contini, D.: Seasonal variability of $PM_{2.5}$ and $PM_{10}$ composition and sources in an urban background site in Southern Italy, Sci. Total. Environ., 612, 202, 2017.

Comero, S., Capitani, L., and Gawlik, B.: Positive Matrix Factorisation (PMF)–An introduction to the chemometric evaluation of environmental monitoring data using PMF, Office for Official Publications of the European Communities, Luxembourg, 59, 10.2788/2497, 2009.

3. What were the limits of detection for the different compounds.

**Response:** The authors have added limit of detection (LOD) of the target compounds **in SI. See Table S1**.

4. Line 131-144: I would recommend converting this to a table and including the chemical structures. I would also consider including the vapor pressures of the different compounds if these have been reported in previous literature (see comment below).

**Response:** Based on all reviewer suggestions, the authors give the chemical structures, the vapor pressures and LOD of these species **in SI. See Table S1**.

**Results:**

5. In the introduction, it is discussed that PAHs are semi-volatile. Can the authors comment on the implications of this on the measured PAH concentrations? Are the differences seen between the heating period and the before heating period driven only by differences in emissions between the two periods, or is it possible that gas-particle partitioning also plays a role.

**Response:** The semi-volatcile nature of PAHs indeed influences their gas-particle partitioning. Lower temperatures (e.g., during the heating period) typically favor partitioning of semi-volatile PAHs into the particle phase, while higher temperatures may enhance volatilization to the gas phase, reducing particle-phase concentrations. Notably, the observed PAH concentration trends aligned more closely with source-specific tracers (e.g., $SO_2$ for coal combustion) during heating than before heating, suggesting that emission variations played an important role. While we acknowledge

that other factors (e.g., lower boundary layer height, meteorological conditions) may amplify the observed differences. Therefore, the authors have added to account for the combined effects of other factors in the analysis of concentration changes in the revised manuscript. Such as "…and this diurnal variation is strongly influenced by the emission sources (e.g. coal and biomass burning, heavy truck traffic), in addition to air temperature and boundary layer height" (**lines 261-264**), and "Like other pollutants, this diurnal variation is influenced not only by the boundary layer height and temperature, but also by the emission source" (**lines 279-281**). While the current assessment of temperature and boundary layer height impacts remains qualitative, we plan to incorporate model-based simulations in future studies to quantitatively disentangle their relative contributions.

6. Line 320-323 states that emissions from coal combustion increases during the heating period, however, line 350 states and figure 6 shows that there is little change in the contribution from coal combustion. Please clarify this contradiction.

**Response:** The authors are sorry to cause such trouble to the reviewer. The sentence "biomass burning emerged as a major source of pollution throughout the entire campaign, while the contribution mediated by coal combustion increased remarkably during the heating period" (**line 355-358**) were mainly qualitatively expressed the absolute contributions of two major sources (biomass combustion and coal burning), based on the relationship between aromatic species and tracer pollutants. And the sentence "there were no significant changes to the relative contributions prior to and during the heating phase" (**line 390-394**) mainly means that the relative contribution of coal combustion source does not change significantly before and during heating. However, since the absolute concentrations of PAHs and NPs increase significantly during the heating period, the actual contribution of coal combustion source also increases. In order to avoid similar misunderstandings, the authors have added instructions as follows: "But the actual contribution of coal combustion source also increases, along with the absolute concentrations of PAHs and NPs increasing during the heating period. Furthermore, coal combustion was also proved an important source of PAHs based on the IP/BghiP and BghiP/BeP ratios (**Table S3**) (Ohura et al., 2004; Grimmer et al., 1983)". **See line 390-396**.

**References**

Grimmer, G., Jacob, J., and Noujack, K. W.: Profile of the polycyclic aromatic hydrocarbons from lubricating oils inventory by GCGC/MS-PAH in environmental materials, Part 1, Fresenius Zeitschrift fur Analytishe Chemie, 314, 13–19, 1983.

Ohura, T., Amagai, T., Fusaya, M., and Matsushita, H.: Polycyclic aromatic hydrocarbons in indoor and outdoor environments and factors affecting their concentrations, Environ. Sci. Technol., 38, 77–83, 2004.

7. I˒m assuming that 4-nitrophenol and 4M5NC were selected for the detailed quantum chemical analysis as they showed the highest contributions. Clarifying if this

was the case would help motivate this section.
**Response:** Suggestion taken. The authors have revised the relevant description, i.e. "Therefore, to better comprehend the formation mechanism of the dominant NPs species in this observation, DFT calculations at the M06-2X/6-311++G(2df,2p) level of theory was carried out to explore the formation mechanism of 4NP and 4M5NC through oxidation processes of the major precursors by OH, $NO_2$ and $NO_3$". **See lines 437-441**.

8. At line 411, the authors state that only the addition at the para- position was considered for mechanisms 1 and 2, why was the ortho- position not considered? In connection with this, 2-nitrophenol concentrations are not reported. Is this because it was not detectable by the method used, or because the concentrations were below the limits of detection. If the quantum calculations show that the mechanism greatly favors the para- substitution, this would support that.
**Response:** The authors have added the reasons as follows: "In the case where the -OH and -$NO_2$ functional groups in 4NP are at the para-position, hence only the *para*-H in phenol and nitrobenzene were chosen for the formation of 4NP". **See lines 459-461**. The quantum chemical calculations in this study focused specifically on the dominant species identified through field observations, i.e. 4NP and 4M5NC. The exclusion of 2-nitrophenol (2NP) from our analysis stems primarily from the unavailability of analytical standards during the experimental phase, which precluded reliable quantitative determination of 2NP in particulate matter. Moving forward, we intend to prioritize the development of analytical methodologies that enable comprehensive characterization of both known and unidentified species through integrated qualitative and quantitative approaches.

9. Similarly, in mechanism 3, is it possible for the nitro group to add first forming an isomer?
**Response:** Suggestion taken. It is possible for the nitro group to add first forming an isomer in other subordinate pathways. To provide a clearer understanding of the different pathways in the reaction mechanism of *P*-Cresol with OH, $NO_2$ or $NO_3$ to the readership, we have drawn a comprehensive figure containing possible parallel pathways and included it in the **SI (Fig. S2)**. As can be seen, there are two parallel pathways leading to the formation of 4M5NC. One involves the initial addition of an OH radical on C2 of the benzene ring, followed by the addition of $NO_2$ on C5 which has been expressed as Mechanism 3 in the manuscript. While the other one involves the initial addition of a $NO_2$ on C5 of the benzene ring, followed by the addition of OH on C2. Based on the calculated reaction barrier and the reactant monomer concentrations, the OH radical induced H-abstraction step possesses a much lower barrier, resulting in the following OH addition step more feasible. Hence, the mechanism 3 may be the dominant pathway and has been shown in the manuscript (**Fig. 8**). An explanation has also been added in **lines 501-510** of the revised manuscript. "A comprehensive reaction mechanism of *P*-Cresol with OH, $NO_2$, $NO_3$ have been drawn to display the potentially different pathways. As can be seen, there

are two parallel pathways leading to the formation of 4M5NC. One involves the initial addition of an OH radical on C2 of the benzene ring, followed by the addition of NO2 on C5 mentioned above. While the other one involves the initial addition of a NO2 on C5 of the benzene ring, followed by the addition of OH on C2. Based on the calculated reaction barrier and the reactant monomer concentrations, the OH radical induced H-abstraction step possesses a much lower barrier, resulting in the following OH addition step more feasible. Hence, the Mechanism 3 may be the dominant formation pathway of 4M5NC".

[Figure]

Fig.S2 Potential reaction mechanism of *P*-Cresol with OH, NO2 or NO3 at the M06-2X/6-311++G(2df,2p) level of theory.

10. K+ and Levoglucosan are both used as tracers for biomass burning, however, only a weak correlation between the two is seen. Additionally, levoglucosan shows a strong relationship with PAHs while K+ shows a better relationship with OPAHs. Can the authors comment on these differences? Are K+ and levoglucosan emitted by different

types of burning? Does this provide additional insight into the emissions of PAHs and OPAHs?

**Response:** The correlation between $K^+$ and levoglucosan was indeed weak in this study (**Fig. 4**). Thus, the authors selected $Cl^-$ as the inorganic tracer alongside levoglucosan based on empirical evidence derived from our dataset, and added the relative  description as follows: "A strong positive correlation between levoglucosan and $Cl^-$ (r =0.58–0.69) and a weaker correlation with $K^+$ (r = 0.3–0.34) were observed (Fig. 4), with both ions showing elevated concentrations during the heating period, suggests additional combustion activities, e.g. biomass burning. It is worth noting that $Cl^-$ is more robustly associated with levoglucosan, a well-established tracer for biomass burning. Thus $Cl^-$ was prioritized as the inorganic tracer for biomass burning in subsequent analyses (Sec.3.3)". **See Fig. 7, lines 226-232**.

Although this paper does not directly show that there is a better relationship with OPAHs between $K^+$ compared with levoglucosan, it can also be roughly explained the difference between them from the perspective of their source and generation mechanism.  It is generally accepted that $K^+$ derives mainly from the direct release of potassium salts in plants during the burning of biomass and that the intensity of its emission is closely correlated with the temperature of the combustion, for example, high temperatures in the burning stage of an open flame tend to release gaseous $K^+$. Levoglucosan is the pyrolysis product of cellulose/hemicellulose at low temperature and hypoxia, especially in the smoldering stage. PAHs is mainly derived from incomplete combustion of organic matter (such as low-temperature pyrolysis in smoldering stage), which is similar to levoglucosan, so they may share similar emission sources or combustion conditions, resulting in a strong correlation. However, in addition to primary combustion emissions, OPAHs can also be produced secondary by atmospheric oxidation of PAHs. While $K^+$ emission is enhanced in high temperature open flame combustion, fuller combustion may promote the production of oxygenated intermediates such as OPAHs. Therefore, the correlation may be better between $K^+$ and OPAHs.

11. Figure 1: SOR and NOR are not discussed in the text.  Either add a description in the text,  or remove it from the figure.

**Response:** Suggestion taken. The authors have removed it from **Fig. 2 in the revised manuscript**.

12. Figure 2: What is the "Other" category? I don't think this is discussed at all in the text, but it does contribute 40% of aerosol mass.  Is this a measurement between measured PM2.5 and the sum of all the speciated components?

**Response:** The "Others" category refers to the difference between the $PM_{2.5}$ measurements and the sum of all the speciated components. The other components in $PM_{2.5}$ could be heavy metals or currently unknown species maybe from the combustion of coal and others during the campaign. And these large percentages of unknown objects have also been found sometimes in previous reports (as follows).

Li, J., et al., Identification of chemical compositions and sources of atmospheric

aerosols in Xi'an, inland China during two types of haze events, Sci. Total. Environ., 566, 230-237, http://dx.doi.org/10.1016/j.scitotenv.2016.05.057, 2016.

Wang, J., et al., Concentrations and stable carbon isotope compositions of oxalic acid and related SOA in Beijing before, during, and after the 2014 APEC, Atmos. Chem. Phys., 17, 981-992, http://doi.org/doi:10.5194/acp-17-981-2017, 2017.

13. Figure 5: Are the differences between before heating and heating period statistically significant? For example, the OPAH concentration was described as "substantially higher" (line 253) during the heating period, but looking at figure 5, the values appear similar.

**Response:** The authors have added statistical analysis and modified the sentence in the revised manuscript to make it clearer, as follows: "The average total concentration of OPAHs ($\sum$8OPAHs) was determined to be 351 ng·m$^{-3}$ across all the sampling periods. Notably higher concentrations were observed during heating (average 378 ng·m$^{-3}$, range 114–812 ng·m$^{-3}$) compared with those before heating (average 311 ng·m$^{-3}$, range 70–875 ng·m$^{-3}$), though the difference did not reach statistical significance ($p>0.05$)". **See line 270-274**.

14. Figure 6: Please make the top panels larger

**Response:** Suggestion taken. The authors have redrawn **Fig. 7 in the revised manuscript**.

15. Table 2: Please include the units

**Response:** Suggestion taken. The authors have added the units in the caption of **Table 2**.

**Minor comments:**

16. How are the "before heating" and "heating" periods defined and how is cutoff of Nov 14 selected. It would also help to show that cutoff in figure 1, similarly to how it is shown in figure 4.

**Response:** The authors totally agree with the reviewer. As suggested by the reviewer, the authors have added the basis for distinguishing "before heating" from "during heating" as follows: "In northern cities of China, centralized heating generally starts on November 15th each year and continues until March 15th of the following year. Therefore, the entire period in this study was divided into two stages….", as seen in **lines 187-190**. Furthermore, the authors have modified Fig. 2, with adding the time periods of "before heating" and "during heating", as seen in **Fig. 2**.

17. Line 121: There is some ambiguity in the methodology. Are the three 15-minute treatments three separate aliquots that are then recombined?

**Response:** The authors have revised the sentence to "…ultrasonic extraction was performed three times, with each session lasting 15 minutes". **See lines 126-127**.

**Typographical comments:**

18. Line 46: a word is missing

**Response:** The authors are sorry for our careless mistakes, and we have added the word "of" in the first sentence, **as seen in line 47**.

19. Line 64: not sure what "carried out" means. Should this be rephrased as "emitted"?

**Response:** Taking the previous questions into account, the authors have revised the whole sentence as follows: "OPAHs are released directly via the incomplete combustion of fossil fuels, biomass, and other organic materials or formed secondarily through photochemical oxidation of parent PAHs by atmospheric oxidants such as ozone, hydroxyl radicals, and nitrate radicals. **See lines 67-70**.

20. Line 119: Does "membrane" mean "filter"?

**Response:** Yes, the authors have revised "membrane" into "sampled filter", **as seen in line 122**.

21. Line 156: typo of theory

**Response:** The authors are sorry for our careless mistakes. As suggested by the reviewer, it has been corrected in the revised manuscript, **as seen in line 157**.

22. Table 2: typo of levoglucosan

**Response:** The authors are sorry for our careless mistakes. As suggested by the reviewer, it has been corrected in the revised manuscript, **as seen in Table 2**.

23. Fig 4: The pie charts are blurry

**Response:** The authors apologize for causing confusion to the reviewer. The authors have uploaded the high-resolution image in the revised manuscript, **as seen in Fig. 5 in the revised manuscript**.

**Responses to the comments from reviewer #3**

**General Comments**

In this work, the authors measured PAHs, oxygenated PAHs, and nitrophenols in a Chinese city and studied their sources. By analyzing their time series and performing PMF, the authors separated the contributions by residential biomass and coal burning and secondary formation by oxidation. Quantum chemical calculations were also performed to probe the mechanism of oxidation. Overall the manuscript is well written and easy to follow. The measurements are interesting and provide some insights into a group of atmospherically important compounds. I have a couple of major comments and I would recommend publication after considering my suggestions.

**Special Comments**

1. I find that the quantum calculations are unnecessary and provide no new insights. I do not see how they are linked to the atmospheric observations. The calculations shows the free energies of the different pathways and, at best, provide qualitative information about relative importance. Even then, they have nothing to do with the atmospheric observations. I suggest dropping the whole section 3.4. They may fit better with an experimental investigation, rather than field observations.

**Response:** We sincerely appreciate the reviewer's insightful comments regarding the role of quantum chemical calculations in this study. The quantum chemical calculations in this study focused specifically on the dominant species identified through field observations, i.e. 4NP and 4M5NC, which were dominant in the PM$_{2.5}$ samples. The calculations explicitly identify which precursors are kinetically favored to form observed NPs, bridging the gap between the source apportionment and atmospheric chemistry. For example, 4NP formation was traced to two precursors: phenol and nitrobenzene. The calculations revealed that phenol-derived pathways exhibit lower reaction barriers compared to nitrobenzene, suggesting phenol is the dominant precursor under typical atmospheric conditions. This explains why phenol-rich sources (e.g., biomass burning) correlate strongly with NP concentrations in the PMF analysis. Our calculations demonstrated that *P*-Cresol is the primary precursor for 4M5NC via nitration and oxidation pathways. This aligns with field observations, as *P*-Cresol is a known emission tracer from biomass burning and coal combustion.

The study is the first to investigate NP formation mechanisms in a petrochemical hub. Prior work has focused on PAHs and OPAHs, but NPs—especially methylated nitrocatechols like 4M5NC—are understudied despite their health and climate impacts. The results provide mechanistic evidence that methyl-substituted phenols (e.g., *P*-Cresol) are critical precursors, offering a new perspective for emission control strategies. Removing this section would leave the study incomplete, as the formation mechanisms for NPs—a major unresolved question raised in the introduction—would

remain speculative. The quantum calculations are not merely ancillary but integral to explaining the observational data. They provide mechanistic clarity for NPs—a poorly understood class of compounds. We hope this clarification underscores the value of retaining Section 3.4.

Thank you for considering our response. We are open to further revisions to improve the manuscript's clarity.

2. In the analysis, the compounds are lumped into the major groups of PAHs, OPAHs and NPs. I am curious if there are more specific differences within each group. For example, there is a lot of analysis that can be done on PAH ratios, some of which are source specific. Furthermore, within NPs, some phenols are specific to biomass burning (the methoxyphenols), whereas simple phenols may come from more sources. Deeper analysis into specific compounds would be really useful.

**Response:** We sincerely appreciate the reviewer's insightful suggestions regarding the deeper analysis of specific compounds within the PAHs, OPAHs, and NPs groups. As highlighted, compound-specific ratios (e.g., PAH source diagnostic ratios) and functional group variations can indeed enhance source apportionment specificity.

In response to this comment, we have added a detailed analysis of PAH ratios in the revised manuscript (**lines 394-396**), including IP/(IP+BghiP) and BghiP/BeP, which are widely recognized as indicators for distinguishing traffic emissions and coal combustion (Grimmer et al., 1983; Ohura et al., 2004). As follows: "Furthermore, coal combustion was also proved an important source of PAHs based on the IP/BghiP and BghiP/BeP ratios (**Table S3**) (Ohura et al., 2004; Grimmer et al., 1983)". For NPs, the authors emphasize that the overwhelming superiority of 4M5NC demonstrates the important contribution of biomass combustion as follows "At the same time, as the most dominant species during the entire observation period, 4M5NC can also prove that biomass burning is the main source of NPs. Because methyl-nitrocatechols have been proposed as tracers for processed biomass burning aerosol, as further oxidized products of VOCs produced after biomass combustion (Iinuma et al., 2010)" (**lines 383-387**).

Thank you for highlighting this important direction!

**References:**

Grimmer, G., Jacob, J., and Noujack, K. W.: Profile of the polycyclic aromatic hydrocarbons from lubricating oils. Inventory by GC/MS-PAH in environmental materials, Part 1, Fresenius Zeitschrift fur Analytishe Chemie, 314, 13–19, 1983.

Ohura, T., Amagai, T., Fusaya, M., and Matsushita, H.: Polycyclic aromatic hydrocarbons in indoor and outdoor environments and factors affecting their concentrations, Environ. Sci. Technol., 38, 77–83, 2004.

3. Line 21 and throughout: the term "nitrated phenols" is used, but the analytes are really nitrophenols.

**Response:** The authors have made changes as suggested by the reviewers, replacing "nitrated phenols" with "nitrophenols". **See lines 21, 49**.

**4. Line 46: missing "of" in "presence benzene ring"**

**Response:** The authors have added the word "of". **See line 47**.

**5. Line 47: aromatic compounds are not really resistant to decomposition. OH addition to a benzene ring occurs at similar rates as OH abstraction from an alkane.**

**Response:** Thank you for highlighting this important nuance in aromatic compound reactivity. The authors have revised the statement into "Aromatic compounds, characterized by the presence of a benzene ring, are known for their structural stability and distinctive aromatic properties". **See lines 47-48**.

**6. Line 53: PAHs are not just semivolatile. They span a wide range of volatilities, from volatile (naphthalene) to non volatile (BaP).**

**Response:** Thank you for this insightful observation. We fully agree that PAHs encompass a broad spectrum of volatilities, ranging from highly volatile species (e.g., Nap) to non-volatile compounds (BaP). We have revised the text to better reflect the diverse volatility range of PAHs as follows: "PAHs are a class of organic compounds comprised of multiple interconnected aromatic rings, spanning a wide range of volatilities from volatile species to non-volatile high-molecular-weight compounds. They are ubiquitously distributed across various environmental settings." **See lines 54-57**.

**7. Line 64: "OPAHs is carried out..." many errors in this sentence**

**Response:** The authors have revised the sentence to improve grammatical accuracy and enhance technical clarity regarding the sources and formation pathways of OPAHs. As follows: "OPAHs are released directly via the incomplete combustion of fossil fuels, biomass, and other organic materials (Oda et al., 2001; Jakober et al., 2007) or formed secondarily through photochemical oxidation of parent PAHs by atmospheric oxidants such as ozone, hydroxyl radicals, and nitrate radicals (Bandowe et al., 2014; Wang et al., 2011; Lin et al., 2015)." **See lines 67-70**.

**References:**

Oda, J., Nomura, S., Yasuhara, A., and Shibamoto, T.: Mobile sources of atmospheric polycyclic aromatic hydrocarbons in a roadway tunnel, Atmos. Environ., 35, 4819-4827, https://doi.org/10.1016/S1352-2310(01)00262-X, 2001.

Jakober, C. A., Riddle, S. G., Robert, M. A., Destaillats, H., Charles, M. J., Green, P. G., and Kleeman, M. J.: Quinone emissions from gasoline and diesel motor vehicles, Environ. Sci. Technol., 41, 4548-4554, https://doi.org/10.1021/es062967u, 2007.

Bandowe, B. A. M., Lueso, M. G., and Wilcke, W.: Oxygenated polycyclic aromatic hydrocarbons and azaarenes in urban soils: A comparison of a tropical city (Bangkok) with two temperate cities (Bratislava and Gothenburg), Chemosphere, 107, 407-414, https://dx.doi.org/10.1016/j.chemosphere.2014.01.017, 2014.

Wang, W., Jariyasopit, N., Schrlau, J., Jia, Y., Tao, S., Yu, T.-W., Dashwood, R. H., Zhang, W., Wang, X., and Simonich, S. L. M.: Concentration and photochemistry of PAHs, NPAHs, and OPAHs and toxicity of $PM_{2.5}$ during the Beijing Olympic Games, Environ. Sci. Technol., 45, 6887-6895, http://dx.doi.org/10.1021/es201443z, 2011.

Lin, Y., Ma, Y., Qiu, X., Li, R., Fang, Y., Wang, J., Zhu, Y., and Hu, D.: Sources, transformation, and health implications of PAHs and their nitrated, hydroxylated, and oxygenated derivatives in PM2. 5 in Beijing, J. Geophys. Res.-Atmos., 120, 7219-7228, https://doi.org/10.1002/2015JD023628, 2015.

8. Line 67: "nitro" is incomplete. PAHs can react with NO3 and NO2

**Response:** Taking the previous questions into account, the authors have revised the whole sentence as follows: "OPAHs are released directly via the incomplete combustion of fossil fuels, biomass, and other organic materials or formed secondarily through photochemical oxidation of parent PAHs by atmospheric oxidants such as ozone, hydroxyl radicals, and nitrate radicals. **See lines 67-70**.

9. Line 112: it seems like a filter is rectangular, but a diameter (with 2 different numbers) is reported. Perhaps it is not a diameter?

**Response:** Thank you for pointing out this inconsistency. The filter is indeed rectangular in shape with dimensions of 230 mm in length and 180 mm in width, rather than having a circular diameter. The use of "diameter" in our original description was an inaccurate term choice. The authors have revised the filter's rectangular dimensions (180 mm× 230mm) using the proper length/width terminology. **See line 116**.

10. Method Lines 124 – 126: It seems like BSTFA derivatization is used. The main objective for BSTFA is to convert acidic hydrogen (e.g. -OH, -COOH) to trimethylsilyl derivatives for GC analysis. However, other than NPs, none of the other analytes really need BSTFA. Were all the samples derivatized, or just some for NP analysis?

**Response:** Thank you for raising this important question. Yes, BSTFA derivatization was applied to all samples in this study. While some analytes (e.g., PAHs) do not strictly require derivatization for GC analysis, many of our target compounds, including those containing -OH, -COOH, and other acidic functional groups— significantly benefit from BSTFA treatment to enhance their volatility, thermal stability, and chromatographic detectability, such as sugars, sugar-alcohols, and dicarboxylic acids, which were also of interest but was not the focus of this present article.

To ensure methodological consistency and avoid potential biases introduced by separate processing workflows, we opted to derivatize all samples uniformly. This approach simplifies the procedure, minimizes experimental variability, and guarantees the comprehensive detection of both derivatization-dependent and non-dependent analytes within a single analytical run. Such treatments have been published in many previous studies (Ren et al., 2021, 2024; Wang et al., 2012).

**References:**

Ren, Y., et al., Non-negligible secondary contribution to brown carbon in autumn and winter: inspiration from particulate nitrated and oxygenated aromatic compounds in urban Beijing. Atmospheric Chemistry and Physics, 2024. 24(11): 6525-6538.

Ren, Y., et al., 2021. Chemical composition of fine organic aerosols during a moderate pollution event in summertime in Beijing: Combined effect of primary emission and secondary formation. Atmos. Environ. 246, 118167.

Wang, G., et al., 2012. Observation of atmospheric aerosols at Mt. Hua and Mt. Tai in central and east China during spring 2009-Part 2: impact of dust storm on organic aerosol composition and size distribution. Atmos. Chem. Phys. 12, 4065-4080.

11. Line 129: Curious about the choice of tridecane as internal standard. It is quite different from the analytes, because it is nonpolar and contains no aromatic groups.

**Response:** The internal standard method was used to calculate the concentration of the target substance, and the detailed quantification method was as followed (take PAHs as an example): The internal standards with known concentrations were added into the standard solutions of PAHs and the sample solutions. The peak area and concentration were defined as A and C, respectively. The ratio of the $A_{PAHs}$ /$C_{PAHs}$ to the ratio of $A_{C13}$ /$C_{C13}$ in the standard solutions of PAHs should be equal to that in the sample solutions. According to the peak area of PAHs, the concentrations of each PAHs in sample solutions could be calculated. The pretreatment method in this study was used to qualitatively and quantitatively analyze many kinds of organic matters as well as PAHs, such as including sugars, sugar-alcohols, dicarboxylic acids and so on, and relevant research results are being prepared. Such treatments have been published in many previous studies (Ren et al., 2021; Wang et al., 2006, 2012, 2015).

**References:**

Ren, Y., et al., 2021. Chemical composition of fine organic aerosols during a moderate pollution event in summertime in Beijing: Combined effect of primary emission and secondary formation. Atmos. Environ. 246, 118167.

Wang G, Kawamura K, Lee S, et al. Molecular, Seasonal, and Spatial Distributions of Organic Aerosols from Fourteen Chinese Cities. Environ Sci Technol, 2006, 40(15): 4619-4625.

Wang, G., et al., 2012. Observation of atmospheric aerosols at Mt. Hua and Mt. Tai in central and east China during spring 2009-Part 2: impact of dust storm on organic aerosol composition and size distribution. Atmos. Chem. Phys. 12, 4065-4080.

Wang G., et al., 2015. Field observation on secondary organic aerosols during Asian dust storm periods: Formation mechanism of oxalic acid and related compounds on dust surface. Atmos. Environ. 113. 169-176.

12. Method: since quartz filters are used, what are the procedures to quantify the positive and negative artifacts for semivolatile organic compounds?

**Response:** Thank you for highlighting this critical methodological consideration. In this study, quartz filters were employed to collect particulate matter samples for the analysis of organic compounds, including some SVOCs. We acknowledge the potential for both positive artifacts (adsorption of gaseous-phase organics onto the filter) and negative artifacts (volatilization losses of particulate-phase SVOCs during sampling or storage) when using quartz filters. To address these artifacts systematically, the following procedures were implemented: On the one hand, field blanks were routinely processed alongside samples to account for any background

contamination introduced during filter handling, storage, or transport. On the other hand, filters were stored immediately blow 0°C after collection to preserve SVOC stability and prevent further evaporation.

13. Line 174 – 175: it may be useful to report the blank values (e.g. <1 ng/m3), or better yet, limits of detection and quantification
**Response:** We appreciate the reviewer's constructive suggestion. In this study, the blank sample analysis showed no serious contamination (less than 5% of real samples), and data reported here were all corrected for the blanks. The authors have added limit of detection (LOD) and concentration in blank samples of the target compounds in the SI. **See lines 177-183, Table S1 in the SI**.

14. Line 179 – 180: it might be best to expand the term "heating". Maybe "residential heating"? Why is Nov 14/15 the cutoff for before and during heating? Was there an official government policy? Or was this determined based on observed SO2 (as detailed later in the paragraph)?
**Response:** As suggested by the reviewer, the authors have added the basis for distinguishing "before heating" from "during heating" as follows: "In northern cities of China, centralized heating generally starts on November 15th each year and continues until March 15th of the following year. Therefore, the entire period in this study was divided into two stages….", **as seen in lines 187-190**.

15. Line 201: Why was 1.6 chosen as the OM/OC ratio? Is there a reference?
**Response:** Suggestion taken. The authors have added the citation (**see line 211**) as follows:
Turpin, B. J. and Lim, H. J.: Species Contributions to $PM_{2.5}$ Mass Concentrations: Revisiting Common Assumptions for Estimating Organic Mass, Aerosol Science & Technology, 35:1, 602-610, https://doi.org/10.1080/02786820119445, 2001.

16. Table 1 shows N = 74. Presumably this means 74 days? So are the measurements reported in this table daily averages?
**Response:** The values in the table represent the mean, minimum, and maximum values for each process. The authors have added the data description (i.g. Mean (Min-Max)) and comments at the bottom of the tables. As follows: [a] *N*: the number of samples. **See Table 1**.

17. Table 1: what does NA under NO mean? Below detection limit?
**Response:** The authors have added the data description at the bottom of the tables. As follows: [b] *NA*: not available. **See Table 1**.

18. Figure 2: Interesting to see that "others" represent about 40% of the PM2.5 mass. That seems rather high to me. Is the mass fraction of metals expected to be high?
**Response:** The "Others" category refers to the difference between the $PM_{2.5}$ measurements and the sum of all the speciated components. The other components in

PM$_{2.5}$ could be heavy metals or currently unknown species maybe from the combustion of coal and others during the campaign. And these large percentages of unknown objects have also been found sometimes in previous reports (as follows).

Li, J., Wang, G., Ren, Y., et al., Identification of chemical compositions and sources of atmospheric aerosols in Xi'an, inland China during two types of haze events, Sci. Total. Environ., 566, 230-237, http://dx.doi.org/10.1016/j.scitotenv.2016.05.057, 2016. Wang, J., Wang, G., Gao, J., et al., Concentrations and stable carbon isotope compositions of oxalic acid and related SOA in Beijing before, during, and after the 2014 APEC, Atmos. Chem. Phys., 17, 981-992, http://doi.org/doi:10.5194/acp-17-981-2017, 2017.

19. Figure 3: I am not sure how the trends show a linear relationship. There seems to be a considerable amount of scatter around the linear regression.

**Response:** Thank you for your valuable comment regarding the scatter observed in the linear regression plots. We fully acknowledge that there is a noticeable scatter around the linear regression lines in the plots. This scatter is indeed a reflection of the inherent variability in atmospheric measurements, which can be influenced by multiple factors such as source variability, meteorological conditions, and measurement uncertainties. Despite the scatter, the linear regression models were chosen because they provide a statistically significant representation of the overall trends between the variables. The *p*-values (p < 0.01 or p < 0.05) indicate that the observed trends are statistically significant, and the correlation coefficients (r = 0.30–0.34) suggest moderate to moderately strong relationships. Moreover, linear regression is a commonly used method in atmospheric science to identify and quantify trends, even in the presence of variability.

20. Line 243: what are the +/- values reported here? Are they standard deviations? It is incorrect to report an uncertainty larger than the measurement itself, as there is zero probability that a concentration can be negative. A standard deviation is useful as an uncertainty only when a measurement follows a normal distribution, and atmospheric measurements do not. It is preferable to report the range of measurements within a percentile range (e.g. 5$^{th}$ to 95$^{th}$ percentile).

**Response:** Thank you for your thoughtful critique and for emphasizing the importance of statistically appropriate uncertainty representation in atmospheric measurements. We agree that percentile-based ranges are more suitable for non-normally distributed data, as they avoid non-physical interpretations (e.g., negative concentrations) and better reflect the variability in skewed datasets.

The authors have deleted all the values of standard deviation in the revised manuscript, but kept the minimum and maximum values, which min-max ranges are used to describe variability in preliminary analyses. Of course, the authors show a percentile range of the relevant data in the figure. **See lines 194-197, 203, 257-259, 298-300, Table 1, Table 2, and Fig. 6**.

21. Line 249: replace "accounting" with "accounting for"

**Response:** Suggestion taken. The authors have added the word "of". **See lines 264-265**.

22. Line 254 – 255: is the difference statistically significant?

**Response:** The authors have added statistical analysis and modified the sentence in the revised manuscript to make it clearer, as follows: "The average total concentration of OPAHs ($\sum$8OPAHs) was determined to be 351 ng·m$^{-3}$ across all the sampling periods. Notably higher concentrations were observed during heating (average: 378 ng·m$^{-3}$, range: 114–812 ng·m$^{-3}$) compared with those before heating (average: 311 ng·m$^{-3}$, range: 70–875 ng·m$^{-3}$), though the difference did not reach statistical significance (p>0.05)". **See lines 270-274**.

23. Line 264: I suggest replacing "foreign sites" with "sites outside of China". Furthermore, the measurements cited are all in Europe.

**Response:** Suggestion taken. The authors have revised "foreign sites" into "sites outside of China". **See line 287**.

24. Lines 272 – 274: again, is the difference statistically significant?

**Response:** The authors have added statistical analysis and modified the sentence in the revised manuscript to make it clearer, as follows: "During the heating phase, the average concentration of $\sum$9NPs was 76 ng m$^{-3}$ (range: 23-175 ng m$^{-3}$), representing a 1.2-fold increase compared with before heating (average: 65 ng m$^{-3}$, range: 4.2-149 ng m$^{-3}$), although this difference was not statistically significant (p>0.05)". **See lines 298-301**.

25. For Figure 5, panels d, e and f, are the whiskers also representing 25$^{th}$ and 75$^{th}$ percentile?

**Response:** The error bars in (d-f) indicate the standard deviation. The authors have revised the caption of **Fig. 6** in the revised manuscript as follows: "Diurnal variation and component concentrations of PAHs (a, d), OPAHs (b, e), and NPs (c, f) before and during heating. The top and bottom of the vertical line for each box in (a-c) correspond to the 95th and 5th percentiles, respectively, and the top, middle, and bottom horizontal lines of the box mark the 75th, 50th, and 25th percentiles of the data range. The white dot in each box represents the mean value. The error bars in (d-f) indicate the standard deviation.".

26. Line 291: again, not really "international". Maybe just "previous measurements in other regions".

**Response:** Suggestion taken. The authors have revised "international research" into "previous measurements in other regions". **See line 319-320**.

27. Line 315: Why not use K instead of Cl for biomass burning?

**Response:** Thank you for raising this critical question. We acknowledge that K$^{+}$ is indeed a widely used inorganic tracer for biomass burning. However, in this study, we

selected Cl⁻ as the inorganic tracer alongside levoglucosan (a well-established organic tracer) based on empirical evidence derived from our dataset. Specifically, we conducted correlation analyses between inorganic elements ($K^+$ and $Cl^-$) and levoglucosan concentrations. The results revealed that Cl⁻ exhibited a statistically stronger correlation with levoglucosan compared to $K^+$ (e.g., Pearson's r = 0.58 for Cl⁻ vs. 0.30 for $K^+$ during heating, **see Fig. 4**). $K^+$'s weaker correlation may reflect contributions from non-combustion sources (e.g., soil resuspension or fossil fuel emissions). This suggests that Cl⁻ may serve as a more robust inorganic marker for biomass burning under the specific conditions sampled in our study. The authors have added the descriptions in the revised manuscript as follows:

A strong positive correlation between levoglucosan and $Cl^-$ (r =0.58–0.69) and a weaker correlation with $K^+$ (r = 0.3–0.34) were observed (**Fig. 4**), with both ions showing elevated concentrations during the heating period, suggests additional combustion activities, e.g. biomass burning. It is worth noting that $Cl^-$ is more robustly associated with levoglucosan, a well-established tracer for biomass burning. Thus $Cl^-$ was prioritized as the inorganic tracer for biomass burning in subsequent analyses (Sec.3.3). **See lines 226-232**.

28. Line 321 – 323: The evidence for the difference between coal burning and biomass burning is not clear. Why does the author argue biomass burning is important throughout the campaign, while coal burning is increasing? From which measurements can one get to this conclusion?
**Response:** We sincerely appreciate the reviewer's insightful question. The distinction between biomass burning and coal combustion contributions is primarily based on the correlation analysis between target pollutants and their respective source-specific tracers, as detailed below: Levoglucosan is a well-established molecular marker for cellulose combustion (e.g., crop residue or wood burning), while $Cl^-$ enrichment is characteristic of biomass burning emissions. The persistent importance of biomass burning was inferred from the strong correlation ($p<0.001$) observed between the aromatic compounds and biomass burning tracers (levoglucosan and $Cl^-$) across all sampling periods (**See lines 346-350, Fig. 7a**), indicating a continuous influence of biomass burning. The heightened role of coal combustion during heating was evidenced by a significant strengthening of correlations between aromatic compounds and $SO_2$ (a key tracer for coal combustion) specifically in this phase. For instance, the correlation between $SO_2$ and these aromatic compounds was negative before heating, but reached a very significant level ($p<0.001$) of positive correlation during heating (**See lines 351-353, Fig. 7b, c**).

29. Line 350: here the authors argue that the second source factor is coal combustion, and its contribution stays constant throughout the campaign. This is contradictory to statement made in Lines 321 - 323. Some clarification here is needed.
**Response:** The authors are sorry to cause such trouble to the reviewer. The sentence "biomass burning emerged as a major source of pollution throughout the entire campaign, while the contribution mediated by coal combustion increased remarkably

during the heating period" (**lines 355-358**) were mainly qualitatively expressed the absolute contributions of two major sources (biomass combustion and coal burning), based on the relationship between aromatic species and tracer pollutants. And the sentence "There were no significant changes to the relative contributions of coal combustion prior to and during heating compared with those of biomass burning and traffic emission" (**lines 390-392**) mainly means that the relative contribution of coal combustion source does not change significantly before and during heating. However, since the absolute concentrations of PAHs and NPs increase significantly during the heating period, the actual contribution of coal combustion source also increases. To avoid similar misunderstandings, the authors have added instructions as follows: "There were no significant changes to the relative contributions of coal combustion prior to and during heating compared with those of biomass burning and traffic emission. But the actual contribution of coal combustion source also increases, along with the absolute concentrations of PAHs and NPs increasing during the heating period. Furthermore, coal combustion was also proved an important source of PAHs based on the IP/BghiP and BghiP/BeP ratios (Table S2)". **See lines 390-396**.

30. Line 421: what about NO3 in the nighttime? I would expect the dominant sink of aromatic compounds in the night time to be reaction with NO3. Oxidation by NO2 is generally rare, with the exception of heterogeneous oxidation of PAHs by NO2 (to form nitro PAHs).

**Response:** Suggestion taken. We have conducted supplementary calculations about the $NO_3$ induced H-abstraction from the benzene ring (**Fig. 8**). As a result, the reaction barrier is 15.79 kcal/mol which is slightly higher than that of OH radical-induced H-abstraction reaction (11.80 kcal/mol) but significantly lower than that of $NO_2$ induced H-abstraction reaction (40.31 kcal/mol). Even so, the $NO_3$ induced H-abstraction is an endothermic process, making it thermodynamic unfeasible compared to OH radical-induced H-abstraction reaction. **See lines 469-474**.

31. Line 434: same comment. When OH levels are low at night, NO3 should be high enough to be the dominant sink. A simple rate calculation should show the difference. The conclusions are too brief and do not quite fit the ACP guidelines.

**Response:** Suggestion taken. As mentioned in the response to Question 30, although the reaction barrier for $NO_3$ induced H-abstraction step is lower than that of $NO_2$, it is still an endothermic process. Hence, the OH induced H-abstraction step not only exhibits the lowest energy barrier but also represents the sole exothermic process, conferring significant advantages in both kinetic and thermodynamic aspects. According to the computational results, H-abstraction step may primarily take place during daytime when OH radical concentrations are relatively higher, rather than $NO_2$ or $NO_3$ induced H-abstraction. As suggested, to enhance the comprehensiveness of the discussion, we have incorporated the $NO_3$ induced H-abstraction step in **lines 501-510** of the revised manuscript.

32. Having 3 corresponding authors seems entirely unnecessary.

**Response:** Thank you for raising this valid concern regarding the designation of corresponding authors. We fully acknowledge that the inclusion of three corresponding authors may seem unnecessary. However, each of these contributors played distinct and critical roles in the completion of this work:

**Dr. Haijie Zhang** led the quantum chemistry calculations and other data analysis.

**Dr. Yuanyuan Ji** oversaw the experimental design and coordinated the cross-institutional collaboration.

**Prof. Gehui Wang** provided critical theoretical insights and funding acquisition for the project.

This collaborative effort involved researchers from multiple institutions and disciplines, and all three corresponding authors were essential to addressing different aspects of the study. We are open to revising this design if the editorial team deems it necessary, but we believe the current transparently reflects the shared leadership and accountability required for this complex interdisciplinary work.

Thank you for your understanding and for the opportunity to clarify this point.